# Engineering of NEMO as calcium indicators with large dynamics and high sensitivity

Jia Li [1,11], Ziwei Shang[2,11], Jia-Hui Chen[3,11], Wenjia Gu[1,11], Li Yao[2,11], Xin Yang[4], Xiaowen Sun[2], Liuqing Wang[1], Tianlu Wang[5], Siyao Liu[5], Jiajing Li[1], Tingting Hou[6], Dajun Xing [2], Donald L. Gill[7], Jiejie Li[1], Shi-Qiang Wang[6], Lijuan Hou[4], Yubin Zhou[5,8] ✉, Ai-Hui Tang[3,9] ✉, Xiaohui Zhang[2] ✉ & Youjun Wang [1,10] ✉

Genetically encoded calcium indicators (GECIs) are indispensable tools for real-time monitoring of intracellular calcium signals and cellular activities in living organisms. Current GECIs face the challenge of suboptimal peak signal-to-baseline ratio (SBR) with limited resolution for reporting subtle calcium transients. We report herein the development of a suite of calcium sensors, designated NEMO, with fast kinetics and wide dynamic ranges (>100-fold). NEMO indicators report $Ca^{2+}$ transients with peak SBRs around 20-fold larger than the top-of-the-range GCaMP6 series. NEMO sensors further enable the quantification of absolution calcium concentration with ratiometric or photochromic imaging. Compared with GCaMP6s, NEMOs could detect single action potentials in neurons with a peak SBR two times higher and a median peak SBR four times larger in vivo, thereby outperforming most existing state-of-the-art GECIs. Given their high sensitivity and resolution to report intracellular $Ca^{2+}$ signals, NEMO sensors may find broad applications in monitoring neuronal activities and other $Ca^{2+}$-modulated physiological processes in both mammals and plants.

GECIs are indispensable tools for real-time monitoring of $Ca^{2+}$ signals[1] and cellular activities[2]. The SBR ($\Delta F/F_0$), defined as the ratio of the absolute fluorescence ($F$) changes ($F - F_0$, or $\Delta F$) over the basal fluorescence ($F_0$), is a key parameter used to gauge the performance of monocolored GECIs[3]. Efforts have been devoted to generate GECIs with faster kinetics (for example, jGCaMP8 series[4]), but progress toward increased maximal fluorescence change has remained

relatively lagging since the development of GECO and GCaMP6 series approximately 10 years ago[5,6].

With a $Ca^{2+}$-sensing module installed within one fluorescent protein (FP), single-FP-based indicators use $Ca^{2+}$-dependent fluorescence changes to report $Ca^{2+}$ transients. Calmodulin (CaM), together with its target peptide (such as RS20 or M13) is among the most commonly used $Ca^{2+}$ sensing module. Two strategies have been applied to link

[1]Beijing Key Laboratory of Gene Resource and Molecular Development, College of Life Sciences, Beijing Normal University, Beijing, China. [2]State Key Laboratory of Cognitive Neuroscience and Learning, IDG/McGovern Institute for Brain Research, Beijing Normal University, Beijing, China. [3]Hefei National Research Center for Physical Sciences at the Microscale, CAS Key Laboratory of Brain Function and Disease, and Ministry of Education Key Laboratory for Membraneless Organelles and Cellular Dynamics, Division of Life Sciences and Medicine, University of Science and Technology of China, Hefei, China. [4]Exercise Physiology and Neurobiology Laboratory, College of PE and Sports, Beijing Normal University, Beijing, China. [5]Institute of Biosciences and Technology, Texas A&M University, Houston, TX, USA. [6]State Key Laboratory of Membrane Biology College of Life Sciences, Peking University, Beijing, China. [7]Department of Cellular and Molecular Physiology, Pennsylvania State University College of Medicine, Hershey, PA, USA. [8]Department of Translational Medical Sciences, School of Medicine, Texas A&M University, Houston, TX, USA. [9]Institute of Artificial Intelligence, Hefei Comprehensive National Science Center, Hefei, China. [10]Key Laboratory of Cell Proliferation and Regulation Biology, Ministry of Education, College of Life Sciences, Beijing Normal University, Beijing, China. [11]These authors contributed equally: Jia Li, Ziwei Shang, Jia-Hui Chen, Wenjia Gu, Li Yao. ✉e-mail: yubinzhou@tamu.edu; tangah@ustc.edu.cn; xhzhang@bnu.edu.cn; wyoujun@bnu.edu.cn

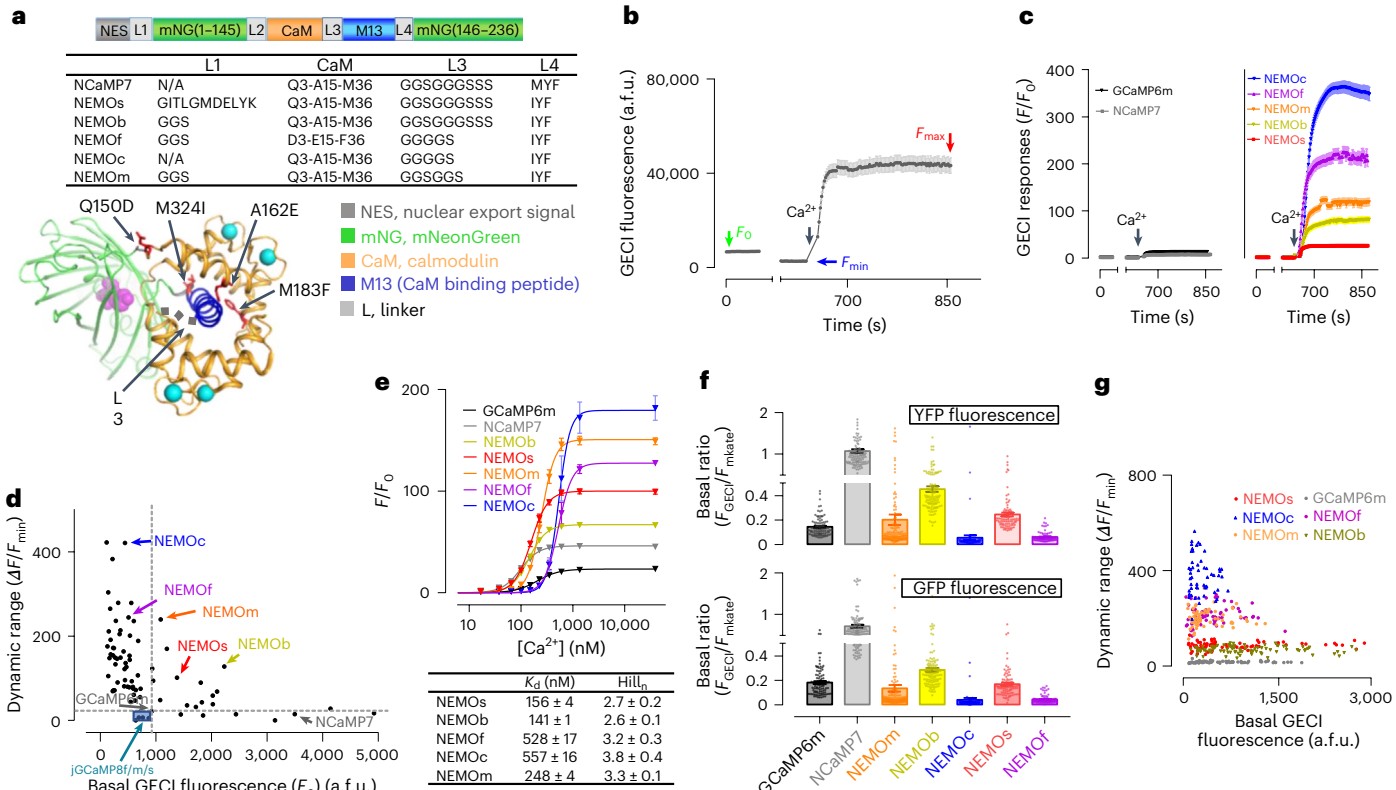

**Fig. 1 | Screening and in vitro characterization of NEMO indicators. a**, NEMO sensors are generated by introducing amino acid substitutions in NCaMP7. Top panel, diagram showing the design of NEMO variants; table (middle panel) or NCaMP7 structure[7] (bottom panel) showing key amino acids substitutions introduced into NCaMP7 to generate NEMO variants. **b–e**, Screening of GCaMP and NCaMP7 variants in HEK293 cells. **b**, $Ca^{2+}$ imaging-based screening. A typical trace from NCaMP7-expressing cells is shown; a.f.u. arbitrary fluorescence units. To avoid saturation of the camera, after recording the basal fluorescence ($F_0$) with regular exposure time (approximately 500 ms), time-series for variants with high dynamic range were recorded using one-tenth to one-fifth the exposure time. Afterwards, the fluorescence response curves of each cell were scaled up according to the corresponding $F_0$. After recording $F_0$, endoplasmic reticulum $Ca^{2+}$ store was depleted using 2.5 μM iono and 1 μM TG. The cells were then incubated in an imaging solution containing 300 μM EGTA, to read minimal GECI fluorescence ($F_{min}$). Finally, the cells were exposed to imaging solution

containing 100 mM $Ca^{2+}$ to obtain the maximal response ($F_{max}$) via SOCE. **c**, Representative traces of GCaMP6m and NCaMP7 (left) or selected NEMO sensors (right). **d**, Scatter plot of $F_0$ – mean dynamic range (DR, $(F_{max} - F_{min})/F_{min}$) of the indicated GECIs. **e**, In vitro dose–response curves of NEMO sensors. $K_d$, dissociation constant for $Ca^{2+}$. Top, typical traces; bottom, statistics (see Supplementary Table 3 for details) ($n$ = 3 independent biological replicates; >17 cells per repeat). Data are shown as mean ± s.e.m. **f**, Basal brightness of NEMO, NCaMP7 or GCaMP6m sensors viewed with YFP (top) or GFP (bottom) filters. To achieve better estimation of the basal fluorescence of GECIs ($F_{GECI}$), $F_{GECI}$ of cells expressing mKate-P2A-GECI constructs was normalized against the fluorescence of mKate, an expression marker ($F_{mKate}$). (GCaMP6m, $n$ = 99 cells; NEMOm, $n$ = 142 cells; NEMOb, $n$ = 119 cells NEMOc, $n$ = 89 cells; NEMOs, $n$ = 91 cells; NEMOf, $n$ = 86 cells; NCaMP7, $n$ = 114 cells). Three independent biological repeats. Data are shown as mean ± s.e.m. **g**, $F_0$ – dynamic range of individual cells expressing NEMO variants or GCaMP6m examined with a YFP filter set.

CaM-M13 with FP: (1) GCaMP-like design[6] to install CaM and M13 to the C terminus and N terminus of an FP; and (2) NCaMP7-like strategy[7] to insert CaM-M13 into the middle of an FP[8]. Modifications within the linkers or interaction interfaces among CaM, M13 and FP were successful strategies to improve the performance of GCaMP variants[4,6,9]. Nevertheless, further improvements in dynamic range (DR) are restricted by the brightness of enhanced green fluorescent protein (EGFP). While NCaMP7 or mNG-GECO[10] was built upon the brightest monomeric green FP, mNeonGreen (mNG)[11], they exhibited a relatively small in cellulo DR[7]. By combining the advantages of both the GCaMP and NCaMP7 series, we set out to develop substantially improved GECIs with fast speed and high DRs building upon mNG.

## Engineering of mNG-based calcium indicators

Single-FP-based indicators share some structural similarities at the sensing module insertion sites[8,12]. Assuming that the design strategies for CaM-based indicators might be transferable in principle among GECIs, we created a series of mNG-based calcium indicator (NEMO) constructs mostly by applying known GECI design

strategies toward mNG, and screened their performance in HEK293 cells (Fig. 1a–c and Supplementary Tables 1 and 2).

We first evaluated their basal fluorescence ($F_0$) and the ratio between maximal ($F_{max}$) and minimal ($F_{min}$) fluorescence, or DR ($(F_{max} - F_{min})/F_{min}$). To allow measurements of $F_{min}$, the endoplasmic reticulum (ER) $Ca^{2+}$ store was depleted by 10-min incubation with 2.5 μM ionophore ionomycin (iono) and 1 μM thapsigargin (TG—an inhibitor of the sarcoplasmic/endoplasmic reticulum $Ca^{2+}$ ATPase). A high amount of $Ca^{2+}$ (100 mM) was added to the bath to induce $F_{max}$ via store-operated $Ca^{2+}$ entry (SOCE) (Fig. 1b). Top candidates (Fig. 1c) were identified based on both the $F_0$ and DR values (Fig. 1d). We found that only the NCaMP7-like[7] design (Fig. 1a) showed improved dynamics and speed (Supplementary Tables 1 and 2). We identified five best-performing constructs and named them as NEMO, including medium (NEMOm), high contrast (NEMOc), fast (NEMOf), bright (NEMOb) and sensitive (NEMOs) versions (Fig. 1d and Extended Data Figs. 1 and 2a).

## Ex vivo characterization of NEMO sensors

The overall in cellulo DR of NEMO sensors is higher compared with that of top-of-the-range GECI proteins tested side-by-side. The DRs

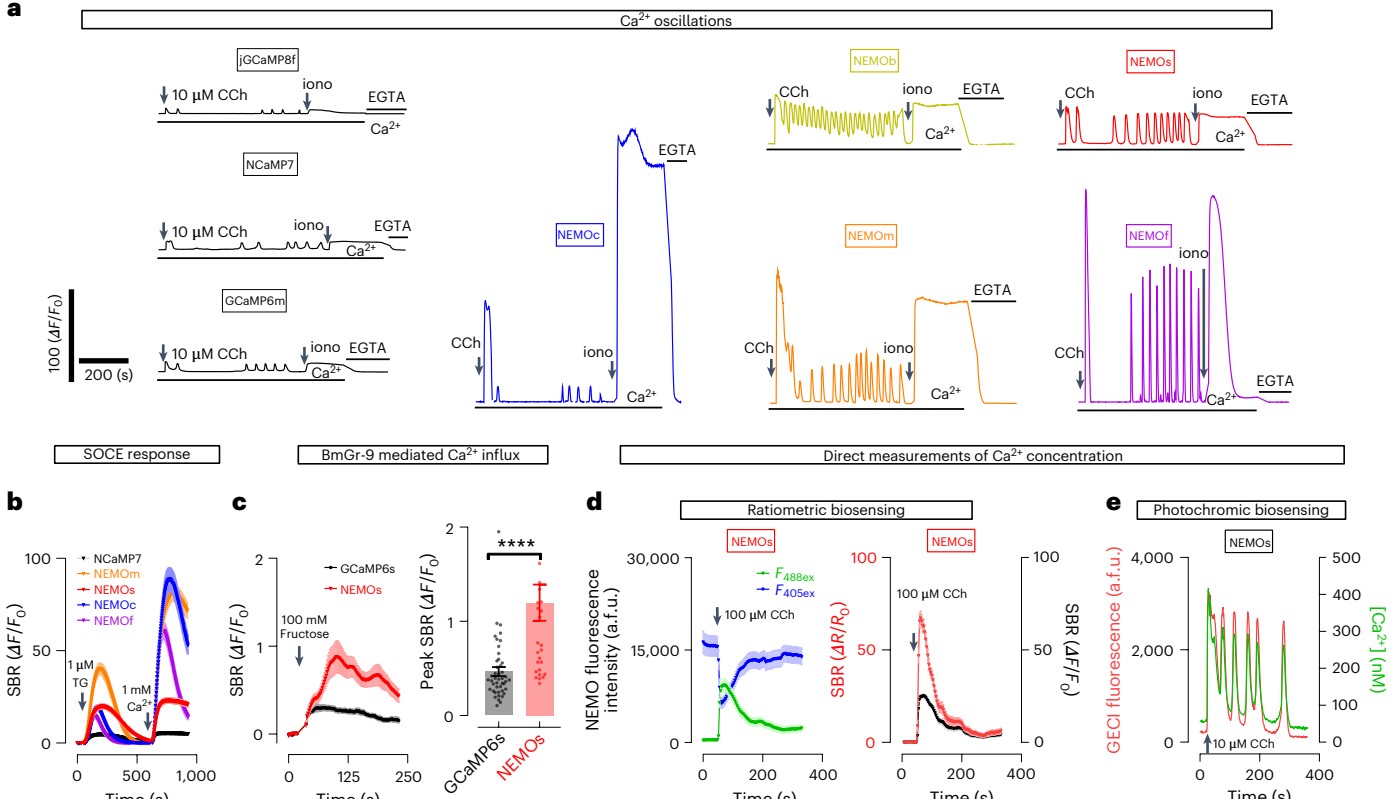

**Fig. 2 | Performance of NEMO sensors in nonexcitable mammalian cells.**
**a**, Typical $Ca^{2+}$ oscillations in HEK293 cells induced by CCh (10 μM), as indicated by GCaMP6m, NCaMP7 and NEMO sensors; $n = 3$ independent biological replicates, with at least 15 cells per repeat. **b**, $Ca^{2+}$ release and SOCE responses induced by TG; $n = 3$ independent biological replicates, with at least 20 cells per repeat. Data are shown as mean ± s.e.m. **c**, Fructose-elicited response in cells coexpressing BmGr-9, an insect fructose receptor. Left, typical traces; right, statistics (****$P = 2.52 \times 10^{-5}$ unpaired Student's *t*-test, two-tailed) (GCaMP6s, $n = 45$ cells; NEMOs, $n = 28$ cells examined over three independent biological replicates). Data are shown as mean ± s.e.m. **d,e**, Measurements of $Ca^{2+}$ concentration with NEMO sensors.

**d**, Ratiometric measurements with NEMOs. NEMO transients in cells upon excitation at 488 nm or 405 nm, induced by 100 μM CCh. Left, typical NEMOs fluorescence response when excited at 488 nm or 405 nm. Right, representative intensiometric (black) or ratiometric (red, $F_{488ex}/F_{405ex}$) responses of the same set of cells shown on the left. $n = 3$ independent biological replicates, with at least 16 cells per repeat. Data are shown as mean ± s.e.m. **e**, iPEAQ of $Ca^{2+}$ levels. Shown are fluorescence intensities (red) and $[Ca^{2+}]$ traces (green) of NEMO-expressing cells in response to 10 μM CCh. In-cell calibration to determine the absolute $Ca^{2+}$ concentration, or $[Ca^{2+}]$, is shown in Extended Data Fig. 7g. $n = 3$ independent biological replicates, with at least nine cells per repeat.

of NEMOs or NEMOb (102.3 ± 4.0 or 128.8 ± 3.1, respectively) were at least 4.5-fold higher than that of GCaMP6m or NCaMP7 (Fig. 1d). And the DRs of NEMOm and NEMOc were further increased to 240.7 ± 7.6 and 422.2 ± 15.3, respectively, 9.5- to 25.7-fold higher than those of GCaMP6m or NCaMP7 (Fig. 1d and Supplementary Video 1).

We further assessed the in vitro performance of NEMO variants. Except for NEMOb, four NEMO sensors showed DR values either close to (NEMOs) or larger than 100-fold (Fig. 1e and Extended Data Fig. 2b). We thus did not focus on NEMOb for further characterization. To our surprise, in vitro DR values were smaller than their corresponding in-cell ones, as GCaMP-like design usually show the opposite[13]. We speculate that macromolecular crowding and reducing condition in cytosolic environment may account for this higher in-cell DR[14,15], which warrants follow-on studies in the future.

We next examined the basal fluorescence of NEMO sensors with a P2A-based bicistronic vector to drive the coexpression of mKate (as an expression marker) and GECIs at a near 1:1 ratio. The basal GECI brightness was indicated by the fluorescence ratio of GECI and mKate (Fig. 1f). Normalized basal brightness of all NEMO sensors was much lower than that of NCaMP7, and the brightness of NEMOc or NEMOf was only around 0.25–0.5 of GCaMP6m. This finding indicates that the lower basal fluorescence of NEMO variants might contribute to the observed large DR of NEMO indicators, in particular for NEMOc

and NEMOf. However, although the DRs of NEMOm, NEMOs and NEMOb were over fivefold higher than that of GCaMP6m, their basal fluorescence was either similar to, or brighter than, that of GCaMP6m (Fig. 1d,f). Hence, high DRs for these three indicators could be attributed to their maximal brightness being larger than that of GCaMP6m as well. In consonance with this notion, NEMO-expressing cells with comparable basal fluorescence to those expressing GCaMP6m still exhibited larger dynamics (Fig. 1g).

Using NEMOc as an example, we next set out to decipher the mechanisms underlying the high DR of NEMO sensors (Extended Data Fig. 3a). Similar to most GECIs[7,16,17], the fluorophore of NEMOc adopted two configurations: an anionic state (peak at 509 nm) and a neutral state (403 nm) (Extended Data Fig. 3b). The $Ca^{2+}$-induced brightening of NEMOc fluorescence is similarly caused by increasing both the proportion and molecular brightness of anionic form (Extended Data Fig. 3c–e and Supplementary Table 4)[7,16,17]. The increase in DR was associated mostly with the considerably dimmer anionic fluorophore of NEMOc (0.22 ± 0.01 $mM^{-1}cm^{-1}$) in the absence of $Ca^{2+}$, which was approximately one-sixth that of NCaMP7 and one-fifth that of GCaMP6m. Compared with GCaMP6m, the high DR of NEMOc was also a result of increased brightness of $Ca^{2+}$-saturated anionic NEMOc (64.26 ± 2.67 $mM^{-1}cm^{-1}$), approximately three times that of GCaMP6m. Moreover, the in vitro and in cellulo DR of NEMOs under two-photon

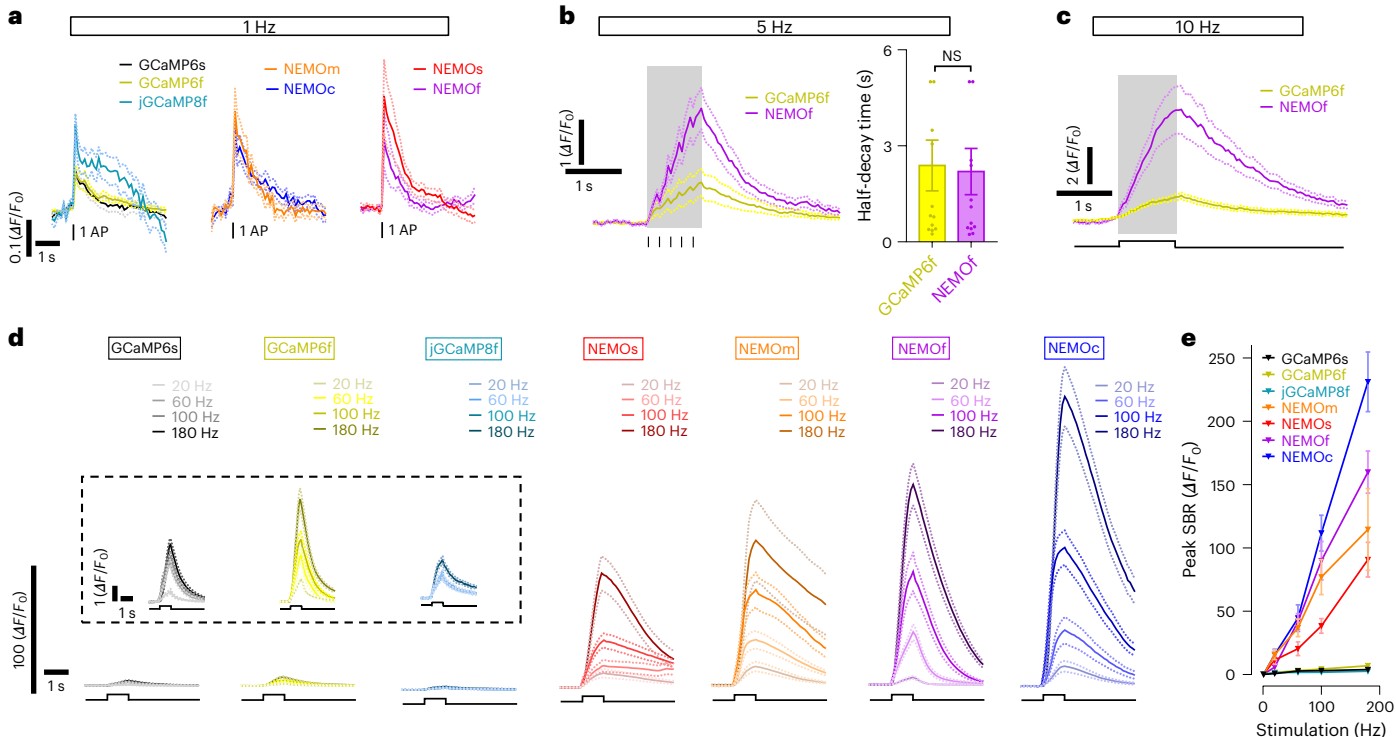

**Fig. 3 | Electric field stimulation-induced NEMO responses in rat hippocampal neurons. a**, Average Ca²⁺ responses reported by GECIs, 1 Hz stimulation (mean traces, $n = 10$–11 cells). **b,c**, Mean NEMOf and GCaMP6f transients induced by 5 Hz (**b**, left, mean traces; right, statistics; $P = 0.86$, unpaired Student's $t$-test, two-tailed, $n = 12$ cells) or 10 Hz (**c**, mean traces, $n = 11$ cells) stimulation. Data are shown as mean ± s.e.m.; NS, nonsignificant. **d**, Mean GECI responses elicited by stimulation at varied frequencies. Inset, enlarged views of responses of reference GECI sensors (SBR magnified by nine

times). **e**, Statistics for data shown in **a** and **d**. Each GECI measurement set was analyzed from several dendrites of at least ten neurons in three different primary hippocampal neuron cultures. All data in this figure are shown as mean ± s.e.m. (for stimulation at varied frequencies, GCaMP6s, $n = 10, 10, 10, 11, 10$ cells; GCaMP6f, $n = 10, 11, 11, 11, 10, 10$ cells; jGCaMP8f, $n = 10, 10, 11, 11, 11$ cells; NEMOm, $n = 10, 12, 12, 13, 12$ cells; NEMOs, $n = 10, 10, 11, 14, 10$ cells; NEMOf, $n = 11, 11, 11, 12, 14$ cells; NEMOc, $n = 11, 11, 11, 13, 12$ cells).

excitation remained largely similar to those observed with one-photon excitation (Extended Data Fig. 4a–c and Supplementary Table 5).

To examine whether it is possible to compensate weaker basal NEMO fluorescence with stronger illumination, we measured the photostability of NEMO sensors. NEMO indicators showed better photostability than GCaMP6m or mNG[11] (Extended Data Fig. 5a, top left two panels). It endured nearly 40 times (1.52 mW) higher illumination than GCaMP6m (0.04 mW), and showed no apparent photobleaching. The stronger illumination (from 0.04 mW to 1.52 mW) could potentially enhance the basal fluorescence of NEMOm sensor by over 60-fold (Extended Data Fig. 5a, top right panel), broadening the applicability of NEMO sensors in scenarios requiring stronger light illumination, such as monitoring Ca²⁺ signals in vivo within subcellular compartments with dim NEMOf indicator.

## Performance of NEMO sensors in nonexcitable cells

We next examined the ability of NEMO sensors to report signals induced by submaximal activation of muscarinic acetylcholinergic receptors with carbachol (CCh, 10 μM). As to the CCh-induced Ca²⁺ transients in HEK293 cells, peak signal-to-baseline (SBR, $\Delta F/F_0$) values of NEMOb and NEMOs were at least three times higher than those of GCaMP6m, jGCaMP8f (ref. 4) and NCaMP7 (Fig. 2a and Extended Data Fig. 5b,c). The SBRs of NEMOm ($101.9 \pm 6.6$), NEMOc ($112.0 \pm 9.8$) and NEMOf ($194.3 \pm 7.7$) were 13–25 times higher than that of GCaMP6m. Similarly, the performance of NEMO sensors in reporting subsequent Ca²⁺ oscillations were also superior to those of jGCaMP8f and NCaMP7 (Fig. 2a, Extended Data Fig. 5b,c and Supplementary Video 2).

We further examined the performance of NEMO sensors in detecting weak Ca²⁺ signals in HEK293 cells. In response to TG-induced Ca²⁺ releases and SOCE, the peak SBR values of NEMO sensors were at least five times higher than that of NCaMP7 (Fig. 2b and Extended Data Fig. 5d). When monitoring Ca²⁺ transients induced by the *Bombyx mori* gustatory receptor (BmGr-9), whose amplitude is much smaller than that of SOCE[18], the responses of NEMO sensors were much stronger than that of GCaMP6s (Fig. 2c).

We next examined whether the larger DR of NEMO sensors could enable more sensitive discrimination of Ca²⁺ signals with varying amplitudes by comparing their performance with GECIs bearing comparable Ca²⁺-binding affinities. We first examined the responses of GECIs responses to the stepwise increase in Ca²⁺ influx induced by a coexpressed optogenetic tool, Opto-CRAC, when subjected to varying photoactivation duration[19–21]. Compared with the GCaMP6m signals, the NEMOm signals were significantly larger (NEMOm, 1,000 ms versus 300 ms, $P = 4.3 \times 10^{-7}$; GCaMP6m, 1,000 ms versus 300 ms, $P = 0.0178$; paired Student's $t$-test, two-tailed), showing a stepwise increase in response to prolonged photostimulation (Extended Data Fig. 6a and Supplementary Video 3). Second, we compared the graded SOCE signals in responses to increased extracellular Ca²⁺ concentrations. NEMOm or NEMOs could discriminate more external Ca²⁺ gradients than did GCaMP6m or NCaMP7 (Extended Data Fig. 6b,c), with SNRs significantly higher than their corresponding counterparts (Extended Data Fig. 6d, 100 ms, $P = 2 \times 10^{-15}$; 300 ms, $P = 1.03 \times 10^{-22}$; 1,000 ms, $P = 5.06 \times 10^{-18}$; Extended Data Fig. 6e, 0.1 mM, $P = 6.95 \times 10^{-23}$; 0.3 mM, $P = 7.01 \times 10^{-26}$; 1 mM, $P = 3.79 \times 10^{-36}$; 3 mM, $P = 1.56 \times 10^{-36}$; 10 mM, $P = 5.85 \times 10^{-37}$; Extended Data Fig. 6f, 0.1 mM, $P = 1.13 \times 10^{-13}$; 0.3 mM,

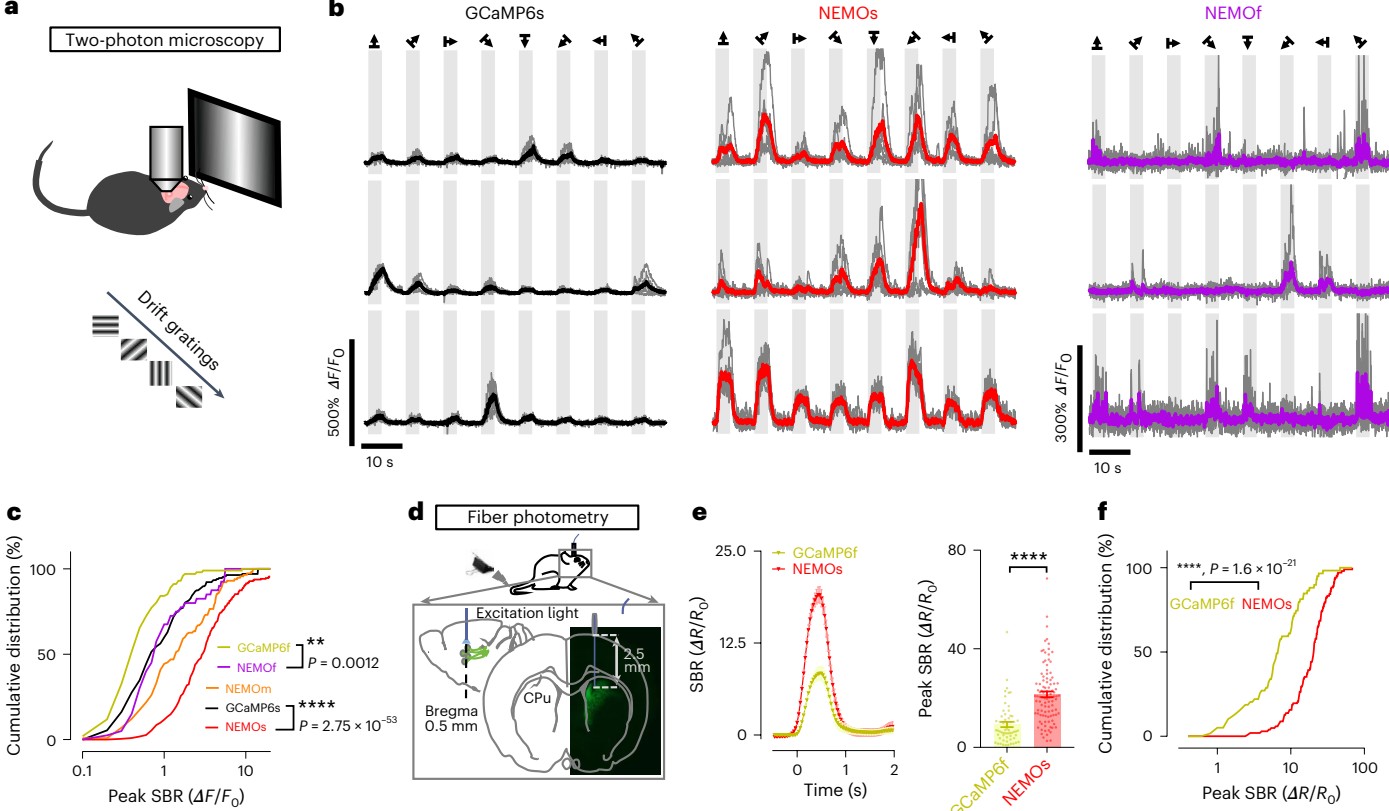

**Fig. 4 | In vivo performance of NEMO sensors in monitoring neuronal activities in rodent brain. a–c,** Fluorescence responses in the visual cortex of mice induced by a visual stimulus. **a,** Diagram showing the experimental setups for two-photon imaging of neurons in response to drift gratings. **b,** Typical response curves for GCaMP6s, NEMOs and NEMOf. **c,** Cumulative distribution of peak SBR transients of GECI sensors (for NEMOf versus GCaMP6f and NEMOs versus GCaMP6s, **$P = 0.012$; ****$P = 2.75 \times 10^{-53}$, Kolmogorov–Smirnov test, two-tailed) (GCaMP6f, $n = 101$ cells from two mice; GCaMP6s, $n = 167$ cells from three mice; NEMOs, $n = 223$ cells from four mice; NEMOm, $n = 115$ cells from

three mice; NEMOf, $n = 40$ cells from three mice). **d–f,** Ratiometric responses of GCaMP6f and NEMOs in neurons of the mouse corpus striatum recorded by fiber photometry. **d,** Diagram of the experimental setup for fiber photometry recordings. **e,** Mean ratiometric responses elicited by pinch stimulation at the mouse tail tip. Left, mean traces; right, statistics (****$P = 4.57 \times 10^{-11}$, unpaired Student's $t$-test, two-tailed). NEMOs, data for 102 cells from six mice; GCaMP6f, data for 59 cells from six mice. Data are shown as mean ± s.e.m. **f,** Cumulative distribution of peak responses shown in **e** (****$P = 1.6 \times 10^{-21}$; Kolmogorov–Smirnov test, two-tailed).

---

$P = 5.13 \times 10^{-16}$; 0.6 mM, $P = 8.85 \times 10^{-20}$; 1 mM, $P = 1.44 \times 10^{-21}$; 3 mM, $P = 3.58 \times 10^{-22}$; unpaired Student's $t$-test, two-tailed).

One major drawback of intensiometric Ca²⁺ sensors is that they could not report Ca²⁺ concentration directly. We thus asked whether NEMO could serve as ratiometric sensors to measure absolute Ca²⁺ concentrations. Indeed, the fluorescence of NEMOs excited by 405-nm light ($F_{405}$) was reduced as a function of increasing Ca²⁺ concentration (Extended Data Fig. 7a). Consequently, the in vitro DR indicated by the $F_{490}/F_{405}$ ratio was 3.4-fold higher than that obtained with $F_{490}$ only (Extended Data Fig. 7b). Similarly, the in-cell ratiometric DRs were significantly larger than intensiometric ones ($P = 0.0002$, unpaired Student's $t$-test, two-tailed) (Fig. 2d and Extended Data Fig. 7c,d).

Violet light illumination approximately doubled the fluorescence of NEMO sensors under low Ca²⁺ conditions, indicating the existence of a photochromic effect[22] (Extended Data Fig. 7e). Indeed, further tests showed that brief 405-nm illumination superimposed on 488-nm light could increase NEMOf fluorescence in an inversely Ca²⁺ dependent manner, with the peak fluorescence named as $F_0$. After switching off the violet light, NEMOf quickly relaxed back to its basal state, termed as the minimal fluorescence ($F_{end}$) (Extended Data Fig. 7f). One such photochromic cycle would allow the calculation of its photochromism contrast, defined as $((F_0 - F_{end})/F_0)_{hv}$ (ref. 22). We thus used NEMOs to report Ca²⁺ concentration with the intermittent

photochromism-enabled absolute quantification (iPEAQ) method[22]. Using basal photochromism contrast together with two in vitro Ca²⁺-titration curves (Extended Data Fig. 7g), we quantified CCh-induced Ca²⁺ releases in terms of absolute Ca²⁺ concentration (Fig. 2e).

## Assessing NEMO sensors in neurons and in planta

We next examined the responses of NEMO sensors in dissociated rat neurons excited by electrical field stimulation with a GCaMP-compatible imaging set up (Fig. 3). We observed that all NEMO sensors were able to detect Ca²⁺ signals elicited by a single action potential (AP) (Fig. 3a), with peak SBR approximately twice as high as GCaMP6s or GCaMP6f. Consistent with its in vitro Ca²⁺ dissociation rate ($K_{off}$) value being higher than that of GCaMP6f (Supplementary Table 3), NEMOf was fast enough to discriminate neuronal responses stimulated with a frequency up to 5 Hz (Fig. 3b).

Similar to what we observed in nonexcitable cells, the high DR of NEMO sensors enabled high-resolution detection of Ca²⁺ signals of various amplitudes. In response to a 5 Hz field stimulation, the peak amplitude of NEMOf response was approximately three times that of GCaMP6f, placing NEMOf among the most sensitive and fast GECIs that include XCaMP[23], jGCaMP7 (ref. 9), or jGCaMP8 series[4]. As the stimulus frequency increased, the difference between peak NEMOf and GCaMP6f responses became more pronounced (5 or 22.7 times higher than that of GCaMP6f; Fig. 3c,d), often with their SNRs

significantly larger than their counterparts (20 Hz, $P = 1.05 \times 10^{-4}$; 60 Hz, $P = 2.24 \times 10^{-7}$, 100 Hz, $P = 2.41 \times 10^{-9}$; 180 Hz, $P = 1.24 \times 10^{-6}$, unpaired Student's $t$-test, two-tailed) (Extended Data Fig. 8a). The NEMO responses were also fairly linear with no apparent saturation even up to 180 Hz (Fig. 3e).

Combining two-photon imaging and whole-cell electrophysiological recording, we further tested NEMO sensors in cortical neurons in acutely prepared mouse brain slices (Extended Data Fig. 8b, left panel). Under a whole-cell patch clamp condition, the intracellular environment could be perturbed and the GECI signal was diluted[23]. Despite this caveat, the responses of all NEMO sensors induced by AP occurring at 50 Hz or higher frequencies were significantly higher than those of GCaMP6, with NEMOf exhibiting the fastest kinetics (NEMOf versus GCaMP6s at 50 Hz, $P = 0.00035$; at 100 Hz, $P = 1.4 \times 10^{-5}$, unpaired Student's $t$-test, two-tailed) (Extended Data Fig. 8b). Of note, most fast GECI sensors developed to date[4,23–25] display rather limited DRs. By contrast, NEMOf expressed in both nonexcitable and excitable cells responds fast with high DR.

We also tested the usability of NEMO sensors in detecting subcellular $Ca^{2+}$ signals in the leaves of *Arabidopsis thaliana*. Our super-resolution imaging results showed that NEMOm fused to the plasmodesmata-localized protein 1 could readily report $Ca^{2+}$ oscillations near the plasmodesmata (Extended Data Fig. 9e and Supplementary Video 4)—a structure between plant cells with a diameter about 30–60 nm (ref. 26).

## In vivo performance of NEMO sensors in rodent brains

We next tested the in vivo performance of NEMO sensors with two-photon laser microscopy (Fig. 4a) by measuring GECI responses as readout of neuronal activity evoked by drifting grating stimulus in the primary visual cortex[27]. To ensure direct comparison with GCaMP6 sensors, we used 920 nm light, an excitation wavelength optimized for GCaMP but less ideal for NEMO (980 nm) to excite the GECIs. All indicators reported differential changes of visual stimuli (Fig. 4b and Extended Data Fig. 8c,d). NEMOf was the fastest among all NEMO variants, with the half-decay time ($409 \pm 54$ ms) comparable with that of GCaMP6f ($482 \pm 48$ ms) (Extended Data Fig. 8e).

We then moved on to compare the sensitivity of GECIs in vivo. As to the fraction of responsive cells (Extended Data Fig. 8f), no significant difference between NEMO variants and the corresponding GCaMP6 indicators was detected (NEMOf versus GCaMP6f, $P = 0.1356$; NEMOs versus GCaMP6s, $P = 0.9805$, unpaired Student's $t$-test, two-tailed). However, the cumulative distribution of peak $\Delta F/F_0$ of NEMOm and NEMOs was substantially right-shifted relative to the GCaMP6 signal (Fig. 4c), indicating that NEMOm and NEMOs are more responsive. The median response of NEMOs ($\Delta F/F_0 = 3$) was over four and seven times stronger than that of GCaMP6s and GCaMP6f, respectively. The visual-stimuli-induced response reported by NEMOs was much larger than existing values reported by existing sensitive GECIs[4,9,25,28]. In parallel, the median response of NEMOf ($\Delta F/F_0 = 0.80$) was significantly larger than that of GCaMP6f ($\Delta F/F_0 = 0.44$) ($P = 0.0202$, unpaired Student's $t$-test, two-tailed) (right panel in Fig. 4b versus Extended Data Fig. 8c, left panel), as well as those reported by the known fastest GECIs[4,9,24,25].

In addition, NEMOs showed appreciably better SNR (Extended Data Fig. 8g) and good basal fluorescence in the mouse V1 that was comparable with that of GCaMP6s even under excitation conditions optimized for GCaMP (Extended Data Fig. 9a,b). NEMOf signal obtained with GCaMP set up showed similar SNR to GCaMP6f (Extended Data Fig. 8g). Since the basal NEMOm fluorescence approximately doubled by switching from GCaMP excitation (920 nm) to a NEMO set up (980 nm), NEMOf under optimized illumination (Extended Data Fig. 9c) retained its large SBR and showed higher SNR (Extended Data Fig. 9d). Since GCaMP6s and NCaMP7 were reported to have

similar in vivo SNR[7], it is likely that optimally excited NEMOs (that is, at 980 nm) may exhibit a better SNR than NCaMP7.

Last, we recorded NEMOs responses in sensory neurons deeply buried in the mouse brain using fiber photometry and settings optimized for ratiometric GCaMP recordings[29]. The ratios of GECI fluorescence excited by 410 nm ($F_{410}$) or 470 nm light ($F_{470}$) were used to indicate $Ca^{2+}$ responses of the neurons within the corpus striatum elicited by tail-pinching stimulus (Fig. 4d). Even though the near-UV light (410 nm) excitation reduced the DR of NEMOs and the 470 nm light was not optimal for NEMOs excitation (Extended Data Figs. 3a and 7d), the median peak response of NEMOs was approximately three times that of GCaMP6f (Fig. 4e,f).

## Conclusions

Here, we reported a GECI toolkit with improved photochemical properties. Unlike current indicators that partially sacrifice the dynamic range for improved sensitivity and/or faster kinetics, NEMO variants are fast acting while still retaining superior dynamic ranges to report $Ca^{2+}$ signals. We would like to point out that when measuring indicators with large in-cell DRs (100–300), the minimal fluorescence used for calculating DR has to be kept close to the background readings to avoid saturation of detectors. Hence, inaccurate subtraction of the background intensity might introduce calculation artifacts, resulting in overestimation of in-cell DR values. However, even with the most conserved estimation, both in vitro and in-cell DRs of NEMOc/m/f were >100-fold, much larger than those of GCaMP6. NEMO indicators are more versatile than the most popular GCaMP series, allowing simultaneous imaging with cyan fluorescence, exhibiting higher photostability that can endure substantially stronger illumination, and better resisting pH fluctuation. Overall, the NEMO sensors may serve as the tool-of-choice for monitoring $Ca^{2+}$ dynamics in mammalian cells, tissue or in vivo, as well as in planta.

## Online content

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

## Methods

### Plasmid construction

The coding sequence of NCaMP7 (ref. [7]) was synthesized by BGI Genel-and Scientific. The corresponding mutations or substitutions were all included on primers (Supplementary Table 6). After PCR, fragments of NEMO were reassembled and inserted into a pCDNA3.1(+) expression vector linearized by digestion with restriction enzymes *Bam*HI and *Eco*RI via the Ready-to-Use Seamless Cloning Kit (B632219, Sangon Biotech). All plasmids were confirmed by sequencing.

### Bacterial expression and protein purification

Transetta (DE3) bacteria (Transgene) transformed with pET28a plasmids containing coding sequences of sensors were cultured with 300 mM isopropyl-β-ᴅ-thiogalactoside for 12 h at 20 °C. Pelleted bacterial cells were then suspended in 20 ml buffer 1 (20 mM Tris, 300 mM NaCl and 1 mM imidazole, pH 7.2), and subsequently sonicated. Recombinant proteins were then purified by 1 ml Ni Sepharose (17-5318-01, GE Healthcare). Columns were sequentially washed with 20 ml buffer 1 and 10 ml buffer 2 (20 mM Tris, 500 mM NaCl and 10 mM imidazole, pH 7.2). Purified proteins were then eluted in 5 ml buffer 3 (20 mM Tris, 100 mM NaCl and 300 mM imidazole, pH 7.2)[9].

### In vitro Ca²⁺ titrations and kinetic measurements

Ca²⁺ titration: 50 μg ml⁻¹ GECIs in buffer A (100 mM KCl, 30 mM HEPES, pH 7.2) supplemented with either 10 mM EGTA or 10 mM Ca-EGTA were mixed in various ratios[9]. GCaMP (excitation: 485 ± 5 nm; emission: 510 ± 5 nm) and NEMO (excitation: 490 ± 5 nm; emission: 520 ± 5 nm) fluorescence were measured with a Flexstation 3 microplate reader (Molecular Devices) controlled by SoftMax Pro v.7.x. The Ca²⁺ titrations curves were fit with Prism v.7 using specific binding with Hill slope function.

For measurements of $K_{off}$ (ref. [9]), 50 μg ml⁻¹ GECIs in buffer A containing either 10 μM free Ca²⁺ or 10 mM EGTA were rapidly mixed with 1:1 ratio GCaMP and NEMO (excitation 485 nm, emission 520 nm) fluorescence were measured by POLARstar Omega microplate reader (BMG LABTECH). $K_{off}$ values were calculated in Prism v.7 using single exponential regression.

### Measurement of spectra and quantum yields of GECIs

Emission and excitation spectra of purified GECIs in buffer A were recorded with a FS5 spectrophotometer (Edinburgh Instruments) controlled by Fluoracle. Absorption spectra were recorded with UV2600 spectrophotometer (Shimadzu) controlled by UVprobe. Quantum yields ($\Phi$) were determined with FS5 using 1 cm quartz cuvette. The fluorescence spectra of GECIs with different concentrations were recorded to calculate the corresponding total integrated fluorescence intensities (TIF). Linear regression of TIF minus absorbance curves were used to derive the slopes ($S$) of GECIs. $\Phi$ was then calculated as: $\Phi_{protein} = \Phi_{standard} \times (S_{protein}/S_{standard})$ (ref. [5]). For anionic chromophore 470 nm excitation; reference, fluorescein (fluo) in 0.1 M NaOH ($\varphi = 0.925$)[30]. For neutral chromophore, 405 nm excitation, TOLLES ($\varphi = 0.79$) (ref. [31]).

### Chromophore extinction coefficients

The absorption spectra of 200 μg ml⁻¹ GECIs in buffers (30 mM trisodium citrate, 30 mM borate, with either 10 mM EGTA or 10 mM CaCl₂, with or without 1 mM MgCl₂) with varying pH were measured with Flexstation 3. Chromophore extinction coefficients ($\varepsilon$) values were then obtained with the following calculations[16,17]. Briefly, the corresponding absorbance (OD) in protonated ($OD_N$, peak absorption at around 400 nm) and deprotonated ($OD_A$, peak absorption at around 500 nm) states were first obtained from the absorption spectrum curves. Next, the slope $S$ was calculated via linear regression using $OD_N - OD_A$ values at pH = 7, 7.2, 8 and 9

$$S = \Delta OD_A / \Delta OD_N \quad (1)$$

The relationship between the concentration of a chromophore ($n$), its OD and $\varepsilon$ is:

$$n = OD/\varepsilon \quad (2)$$

For alkaline-denatured (D), protonated (N) and deprotonated (A) chromophores, the corresponding equation would be $n_D = OD_D/\varepsilon_D$, $n_A = OD_A/\varepsilon_A$ and $n_N = OD_N/\varepsilon_N$. And green GECIs will be totally denatured at pH 12.5, thus $n = n_D$. Also $n$ will not change during H⁺ titration, hence

$$n_D = n_A + n_N \quad (3)$$

and

$$\Delta n_A = -\Delta n_N \quad (4)$$

Putting equation (2) or its variants into (3) and (4) would make:

$$OD_D/\varepsilon_D = OD_A/\varepsilon_A + OD_N/\varepsilon_N \quad (5)$$

$$\Delta OD_A/\varepsilon_A = -\Delta OD_N/\varepsilon_N \quad (6)$$

$\varepsilon_D$ of denatured GFP-like chromophores (44,000 M⁻¹ cm⁻¹) was taken as $\varepsilon_D$ of green GECIs at pH 12.5 (ref. [1]).

And equation (6) can be rearranged as:

$$\varepsilon_A/\varepsilon_N = -\Delta OD_A/\Delta OD_N \quad (7)$$

Combining equation (1) and (7), we will obtain

$$\varepsilon_A/\varepsilon_N = -S \quad (8)$$

By solving equations (5) and (8), we can get

$$\varepsilon_N = \frac{\frac{OD_A}{-S} + OD_N}{OD_D/\varepsilon_D} \quad (9)$$

$$\varepsilon_A = \varepsilon_N \times (-S) \quad (10)$$

and

$$\rho_A = \frac{n_A}{n_A + n_N} = \frac{\frac{OD_A}{\varepsilon_A}}{\frac{OD_A}{\varepsilon_A} + \frac{OD_N}{\varepsilon_N}} \quad (11)$$

$$\rho_N = \frac{n_N}{n_A + n_N} = \frac{\frac{OD_N}{\varepsilon_N}}{\frac{OD_A}{\varepsilon_A} + \frac{OD_N}{\varepsilon_N}} \quad (12)$$

The pKa values were calculated with the 'specific-binding-with-Hill-slope' function using Prism v.7.

### Two-photon imaging

This was undertaken with a ZEISS LSM880 NLO microscope equipped with ×40 water-immersion objective (numerical aperture (NA) 1.0), gallium arsenide phosphide photomultiplier tubes and ZEN v.2.1 software.

In vitro excitation spectra: two-photon cross-section ($\delta$) values were determined[32] using the same imaging solutions as those in $\Phi$ measurements.

The specific calculation steps are as follows[23]. The time-averaged fluorescence photon flux < $F(t)$ > is given as:

$$\langle F(t) \rangle \approx \frac{\phi \eta_2 C \delta g_p}{2f\tau} \times \frac{8n \langle P(t) \rangle^2}{\pi \lambda} \quad (13)$$

where $\delta$ of GECIs can be calculated as:

$$\delta \approx \frac{2 \langle F(t) \rangle \pi \lambda f \tau}{\phi g_p 8 \eta_2 C n \langle P(t) \rangle^2} \tag{14}$$

$\Phi$ is the fluorescence collection efficiency of the measurement system; $\eta$ represents the quantum yield; $\eta_2$ is obtained by two-photon excitation and $\eta_2$ derived from one-photon excitation (we set $\eta_1 = \eta_2 = 0.925$ for fluorescein, assuming that $\eta$ was independent of excitation wavelengths. For $\eta_1$ of GECIs, please see Supplementary Table 4). $C$ is the concentration of the sample; $\delta$ is the two-photon cross-section; $g_p$ is the degree of second-order temporal coherence, which depends on the pulse shape; $f$ is the pulse repetition rate; $\tau$ is the temporal pulse width; $n$ is the refractive index of the sample medium ($n = 1.3334$ for water); $\langle P(t) \rangle$ is the time-averaged instantaneous incident power of the laser beam (we consider this as a constant) and $\lambda$ stands for the excitation wavelength.

As $\frac{\phi g_p \times \langle P(t) \rangle^2}{f \tau}$ value is a system-specific constant at a certain wavelength setting, we could obtain $\delta_{GECI}$ ($\delta$ of GECIs) by using fluorescein at 750–990 nm as a standard[33]. According to the above method, we measured the $\langle F(t) \rangle$ values of each GECI and calculated the $\delta_{GECI}$ values of them by equation 15:

$$\delta_{GECI} \approx \frac{\langle F(t) \rangle_{GECI} \times \eta_{2fluo} \times C_{fluo}}{\langle F(t) \rangle_{fluo} \times \eta_{2GECI} \times C_{GECI}} \times \delta_{fluo} \tag{15}$$

Time-lapse $Ca^{2+}$ imaging of HEK293 cells was acquired using 910 nm (for GCaMP6m) or 970 nm (for NCaMP7 or NEMOs) excitation, and the resulting emission between 500 nm and 600 nm was collected every 2.5 s.

## Cell culture and transfection
HEK293 and HeLa cells (ATCC, catalog nos. CRL-1573 and CL-0101, respectively) were cultured regularly in DMEM supplemented with 10% FBS (and 5% penicillin and streptomycin at 37 °C with 5% $CO_2$ (refs. 34,35). Transfection was performed by electroporation using the Bio-Rad Gene Pulser Xcell system. Transfected cells were seeded on round coverslips and cultured in OPTI-MEM medium containing 7% FBS. All experiments were carried out 24 h after transfection.

Hippocampal tissue from E18 Wistar rats was first digested with 0.1% trypsin at 37 °C for 15 min, washed with mixed complete medium (L-glutamine DMEM-F12 plus 10% FBS) and then pipetted with a Pasteur glass tube (15 mm) to get a cell suspension. Neurons were then plated at 50,000 cells per 18 mm coverslips coated with 0.1% poly-D-lysine and cultured in a 37 °C, 5% $CO_2$ incubator. Half of the medium was replaced with neurobasal medium (containing 2% B27 supplement, 0.5 mM GlutaMAX-100X) without serum 24 h later, and once per week thereafter. Arac (10 μM) was added 7 days after plating to inhibit gliacyte. Neurons were transfected with GCaMP6 or NEMO-encoding plasmids at 7–9 days after plating using a calcium phosphate transfection kit (Takara Bio).

## Fluorescence imaging
Time-lapse fluorescence imaging experiments were carried out using a ZEISS observer Z1 imaging system controlled by SlideBook v.6.0.23 (ref. 12). Cells were imaged in $Ca^{2+}$ imaging buffer (107 mM NaCl, 7.2 mM KCl, 1.2 mM $MgCl_2$, 11.5 mM glucose and 20 mM HEPES-NaOH, pH 7.2) every 1 or 2 s. Filters: NEMO/NCaMP7, $500 \pm 12$ nm $_{ex}/542 \pm 13$ nm $_{em}$; GCaMP, $470 \pm 11$ nm$_{ex}/512 \pm 13$ nm$_{em}$.

## Confocal imaging and iPEAQ measurements
Confocal imaging was undertaken with a ZEISS LSM880 microscope equipped with a ×63 oil objective (NA 1.4) and ZEN v.2.1 software. Excitation was set at 488 and 405 nm, with the emission collected at 500–580 nm.

For the iPEAQ biosensing[36], an in vitro $Ca^{2+}$ titration curve was first generated. NEMOs fluorescence in solutions with different free $Ca^{2+}$ concentrations was imaged with the same $Ca^{2+}$ concentrations as those for live-cell imaging. Eight photochromic cycles induced by 2.5 s illumination with 405 nm light were superimposed on 488 nm light to obtain parameters needed to calculate photochromism contrast. The photochromism contrast is defined by peak fluorescence ($F_0$) in the presence of 405 nm light together with 488 nm light, and minimal fluorescence ($F_{end}$) in the presence of 488 nm light only: photochromism contrast = $((F_0 - F_{end})/F_0)_{hv}$, or $(\Delta F/F_0)_{hv}$. Afterwards, dose–response curves (shown as $(\Delta F/F_0)_{hv}$) and NEMOf fluorescence were plotted and fit with a Hill equation.

In-cell iPEAQ recording of NEMOs were similarly imaged with 488 nm laser. First, five photochromic cycles with 2.5 s illumination at 405 nm were superimposed on 488 nm light to obtain basal $(\Delta F/F_0)_{hv}$. Then these values of each single cell were applied to the $(\Delta F/F_0)_{hv}$ $Ca^{2+}$ titration curve to obtain basal $[Ca^{2+}]$. Afterwards, the measured basal NEMOf fluorescence and calculated basal $[Ca^{2+}]$ were applied to the NEMOf fluorescence $Ca^{2+}$ titration curve to obtain the normalizing factor needed to convert in cellulo NEMOf fluorescence to the corresponding in vitro NEMOf fluorescence, and then to real-time $[Ca^{2+}]$[36].

## Light-tunable activation of $Ca^{2+}$ entry in HeLa cells
HeLa cells cotransfected with Opto-CRAC (mCherry-LOV2$_{404–546}$-STIM1$_{336–486}$) and NEMOm or GCaMP6m were plated on glass-bottomed dishes (Cellvin, catalog no. D35-20-0-TOP), imaged using a Nikon Ti2 Inverted microscope with a Yokogawa W1 dual spinning disk scanhead, Micro-Scanner for photostimulation and stage top incubator for live-cell imaging[19]. To prevent preactivation of Opto-CRAC, we acquired $Ca^{2+}$ signals (NEMOm or GCaMP6m) under emission light of 525 nm with 1% excitation strength of 488 nm and 10 ms exposure time. To tune the activation of Opto-CRAC, a Nikon 'A1 Stimulation' toolbar was applied with 488 nm stimulation (2% strength). Varied exposure times were applied in 'A1 Stimulation' to control Opto-CRAC activation and photo-induced $Ca^{2+}$ influx.

## $Ca^{2+}$ imaging in neurons
Neurons of DIV 17-20 were imaged using a W1 spinning disc confocal microscope (Ti2-E, NIKON) with a 100 × oil-immersion objective (1.45 NA, NIKON) in Tyrode's solution (129 mM NaCl, 5 mM KCl, 30 mM glucose, 25 mM HEPES-NaOH, pH 7.4, 1 mM $MgCl_2$ and 2 mM $CaCl_2$). Field stimulations were performed in a stimulation chamber (Warner Instruments, RC-49MFSH) with a programmable stimulator (Master-8, AMPI). Samples were excited with 488 nm laser and fluorescence was collected with a Zyla4.2 sCMOS camera (Andor) by NIS-Elements AR 5.10.00 software.

## Data analysis for $Ca^{2+}$ imaging in cells or cultured neurons
The corresponding mean fluorescence of regions of interest (ROIs) were analyzed by Matlab v.2014a (The MathWorks) and plotted with Prism v.7 software (Matlab codes are available upon request).

## $Ca^{2+}$ imaging and electrophysiology in cortical slices
Cortical slices were prepared from the adult mice (>P50, either of sex), 3 weeks after the stereotaxic injection of various adeno-associated virus (AAV) vectors encoding different $Ca^{2+}$ probers to the primary visual cortex (V1), following a protocol described in our previous studies[37,38]. $Ca^{2+}$ probe-expressing pyramidal cells (PCs) in cortical slices were recorded in a two-photon laser scanning microscopy system (model FV1200MPE, Olympus) equipped with a wavelength-tunable Mai-Tai femtosecond infrared laser (DeepSee, Spectra Physics). Whole-cell current-clamp recording on individual PCs was made with the glass micropipettes, filled with the internal solution containing 130 mM K-gluconate, 20 mM KCl, 10 mM HEPES, 4 mM Mg$_2$ATP, 0.3 mM Na$_2$GTP and 10 mM Na$_2$-phosphocreatine (pH 7.25–7.35, adjusted with

KOH, 305 ± 5 mOsm), and a microelectrode amplifier (Axon MultiClamp 700B, Molecular Devices). Membrane potentials were low-pass filtered at 10 kHz, digitized at 20 kHz (DigiData 1440A, Molecular Devices) and acquired by the pClampex v.10.3. Simultaneously time-lapse imaging of $Ca^{2+}$ fluorescence signals in the soma of recorded PCs, evoked by APs at various frequencies or numbers, was acquired by 17–20 Hz at a 256 × 256 pixel resolution using a ×40 water-immersion objective (LUMPlanFL, NA 0.8, Olympus) and the 920 nm laser wavelength. $Ca^{2+}$ signals for each tested AP trains were averaged by three sweeps. The signals of neuronal AP and $Ca^{2+}$ fluorescence were synchronized with an analog connection unit (FV10-ANALOG, Olympus).

The acquired time-lapse images were analyzed offline with the OLYMPUS FV10-ASW v.4.2 (Olympus) and our custom MATLAB (MathWorks) scripts. A subtraction of the background fluorescence region outside the PC soma was made to estimate the basal fluorescence intensity $F_0$, and the average $F_0$ for a 0.5-s duration before the onset of AP trains was used in the calculation of $\Delta F/F_0$ (ref. [23]). The latter quantification process was the same as that used in a previous study[9].

### In vivo two-photon laser $Ca^{2+}$ imaging of mice V1 neurons

**Animal surgery and virus injection.** The use of animals was approved by the Institutional Animal Care and Use Committees of Beijing Normal University, Peking University and University of Science and Technology of China. Mice at postnatal days 15–20 (P15–P20) were injected transcranially with 700 nl AAV-Syn-GCaMP6f, AAV-Syn-GCaMP6s or AAV-Syn-NEMO virus in the V1. After 3 weeks, animals were used to perform in vivo and ex vivo calcium imaging experiments. A small craniotomy (2.5 mm × 2.5 mm) was made on the V1 area (centered 2.5 mm left, and 0.5 mm anterior to lambda suture) in the mouse under anesthesia ketamine/medetomidine (50 mg kg⁻¹, 0.6 mg kg⁻¹; intraperitoneal)[39] and then covered with a 3-mm diameter round glass-coverslip. A chamber around the craniotomy, made by dental cement, was filled with the Ringer's solution containing: 123 mM NaCl, 1.5 mM CaCl₂, 5 mM KCl. After a 30-min recovery from the surgery, mice were transported to the two-photon-laser imaging set up.

**Visual stimuli.** Visual stimuli were generated by a custom-developed software using LabVIEW v.8.5 (National Instruments) and MATALB (Mathworks), and were presented on a liquid crystal display monitor (ThinkVision, Lenovo) for in vivo calcium imaging experiments[39]. The simulation covered 0° to 80° horizontal visual field and −35° to 40° vertical visual field. Full screen drifting gratings with eight different orientations (spatial frequency, 0.02 Hz per degree; temporal frequency, 2 Hz, 100% contrast) were presented in a pseudorandom order, and each orientation with 4-s duration was assessed six times at intervals of 7–8 s blank gray-screen stimulus (with identical mean luminance).

**Two-photon imaging and $Ca^{2+}$ signal analysis.** Time-lapse $Ca^{2+}$ imaging from the V1 layer 2/3 PCs was conducted with Scanbox v.4.1 system, or a custom-modified Olympus two-photon laser scanning microscopy system (model FV1200MPE). The excitation wavelength was set at 920 nm. Fluorescence signals were acquired at 11–14 Hz. For each acquired fields of view (FOV), ROIs were set manually on visually identified neuronal cell bodies.

$Ca^{2+}$ signal analysis was performed, using our custom MATLAB scripts[9]. $F$ is instantaneous fluorescence, the averaged baseline fluorescence intensity of 1-s duration before visual stimulation onset was calculated as $F_0$, and $Ca^{2+}$ responses were defined as $\Delta F/F_0 = (F - F_0)/F_0$. GECI responses evoked by optimal visual orientation stimulus with $P$ values <0.01 (Student's $t$-test) were identified as responsive cells. For each responsive cell, the peak $\Delta F/F_0$ responses, the peak SNR, and the half-decay time of the maximal $\Delta F/F_0$ responses were calculated, respectively, as follows.

For the half-decay time, an exponential function was used to fit $\Delta F/F_0$ responses of GCaMP6s, NEMOm or NEMOs that were averaged

over six trials with optimal stimulus, while the same function was used for maximal peak $\Delta F/F_0$ responses of GCaMP6f or NEMOf from one trial of optimal stimulus.

The peak SNR was calculated as

$$\text{peak SNR} = \frac{\text{peak } \Delta F/F_0}{\text{SD}_{\text{baseline}}} \tag{16}$$

Where $\text{SD}_{\text{baseline}}$ is the standard deviations of $\Delta F/F_0$ responses before 1-s visual stimulus presentation.

The orientation selectivity index and direction selectivity index were calculated by using mean $\Delta F/F_0$ amplitude (averaging the top 25% of $\Delta F/F_0$ responses during 4-s stimulus presentation) over six trials evoked by individual eight orientation stimulation. The orientation selectivity index was calculated as:

$$\text{OSI} = \frac{\sqrt{\left(\sum_i \left(R\left(\theta_i\right) \times \sin\left(2\theta_i\right)\right)\right)^2 + \left(\sum_i \left(R\left(\theta_i\right) \times \cos\left(2\theta_i\right)\right)\right)^2}}{\sum R\left(\theta_i\right)} \tag{17}$$

where $\theta_i$ is orientation of drifting gratings, $R(\theta_i)$ is the mean $\Delta F/F_0$ amplitude at $\theta_i$.

The direction selectivity index was calculated as

$$\text{DSI} = \frac{R_{\text{pref}} - R_{\text{opp}}}{R_{\text{pref}} + R_{\text{opp}}} \tag{18}$$

Where $R_{\text{opp}}$ is the mean $\Delta F/F_0$ amplitude at the opposite angle to the preferred angle.

### Tail-pinching stimulus and optical fiber recording

After 3 days of adaptive feeding, six male C57BL/6 mice (7 weeks old, 20–25 g) were divided randomly into GCaMP6f group and NEMOs group ($N = 3$). *AAV-Syn-GCaMP6f* or *AAV-Syn-NEMOs* virus were injected into the striatal region (AP, +0.5 mm; $R$, 1.8 mm; DV, −2.5 mm), respectively. Two weeks after transfection, the experiment was performed using the Reward R810 dualcolor multichannel optical fiber recording system controlled by ORFS v.2_14397.

Pinch stimulation at the tail tip was given by a 15-mm long tail clip. Excitation light (410/470 nm; 17.5/65 μw) and emission between 500–550 nm were transferred via an optical fiber implanted into the virus injection area of anesthetized mice fixed on a stereotaxic instrument. GECI signal excited with 410 nm was a $Ca^{2+}$-independent reference to cancel out motion artifacts[40]. Pinch-induced GECI responses were recorded by with a rate of 60 fps. One stimulation was defined as one event. For GCaMP6f, $n = 97$; for NEMOs, $n = 101$.

### Animals

All mice were housed in a 12 h light/dark cycle. Food and water were provided ad libitum. The temperature of the room was controlled at 20–25 °C, and the humidity was maintained at 45–60%.

### Super-resolution imaging of plasmodesmata

Full-length cDNA of plasmodesmata-localized protein 1 (PDLP1) was tagged with NEMOm and cloned into pCAMBIA1390. PDLP1 is a well-established plasmodesmata marker[26]. To obtain transgenic plants, *Agrobacterium* GV3101 containing the resulted construct was transformed into wild-type *Arabidopsis thaliana* using the floral dip method[41]. Images were collected from the abaxial leaves of 10-day-old seedlings at 2-s intervals using an Airyscan LSM880 confocal microscope.

### Reporting summary

Further information on research design is available in the Nature Portfolio Reporting Summary linked to this article.

## Data availability

The coding sequence of NEMO sensors have been deposited with GenBank (NEMOf, OQ626715; NEMOc, OQ626716; NEMOb, OQ626717; NEMOm, OQ626718; NEMOs, OQ626719). Key NEMO plasmids are available via Addgene (189930–189934). Source data are provided with this paper.

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

## Acknowledgements

This work was supported by the National Natural Science Foundation of China (91954205 and 92254301 to Y.W.; 32130043 and 32071025 to X.Z.; 31872759 to A.-H.T.; 31971095 to L.H. and 32171033 to D.X., 91854209, 32230048 to S.-Q.W., 92054101 and 32122013 to J.L.), National Science and Technology Innovation 2030 Grants (2022ZD0204900 to X.Z.; 2021ZD0202503 to A.-H.T.), the National Key Research and Development Program of China (2020YFA0112200 to A.-H.T.), the Ministry of Science and Technology of China (2019YFA0802104 to Y.W.) and the China Postdoctoral Science Foundation (2021M703089 to J.C.).

## Author contributions

Y.W., X.Z., A.-H.T., L.H. and Y.Z. supervised and coordinated the study. J.L. designed and generated all the plasmid constructs. J.L. and W.G. performed the in vitro assays. J.L. performed all fluorescence imaging in HEK cells, with help from W.G. and L.W. T.W. and S.L. performed the optogenetic experiments. J.-H.C. performed confocal imaging of dissociated neurons. T.H. performed confocal imaging of dissociated cardiac cells. L.Y. performed two-photon Ca$^{2+}$ imaging of whole-cell patched neurons in mouse brain slices. J.L. made the transgenetic *Arabidopsis thaliana* and carried out super-resolution imaging of plasmodesmata. Z.S. performed in vivo two-photon Ca$^{2+}$ imaging of visual cortical neurons in mice. X.S. monitored basal fluorescence of indicators in mice visual cortex. X.Y. performed fiber photometry in vivo. J.L., Z.S., J.C., W.G., L.W., L.Y., Z.S., X.Y. and X.S. analyzed data, with input from the other authors. D.L.G., S.-Q.W., D.X. and J.L. provided intellectual input to the manuscript. Y.Z., X.Z. and Y.W. wrote the manuscript with inputs from all the other authors.

## Competing interests

The authors declare no competing interests.

## Additional information

**Extended data** is available for this paper at https://doi.org/10.1038/s41592-023-01852-9.

**Correspondence and requests for materials** should be addressed to Yubin Zhou, Ai-Hui Tang, Xiaohui Zhang or Youjun Wang.

```
                 *         *         *         *         *         *         *         *
                 |         |         |         |         |         |         |         |
NCaMP7 : ------------------------------------MVSKGEEENMASLPATHELHIFGSINGIDFDMVGQGTGNPNDGY :  44
NEMOc  : ----------------------------MHHHHHHVSKGEEENMASLPATHELHIFGSINGIDFDMVGQGTGNPNDGY :  50
NEMOm  : MLQNELALKLAGLDINKTGGGS---------HHHHHHVSKGEEENMASLPATHELHIFGSINGIDFDMVGQGTGNPNDGY :  71
NEMOs  : MLQNELALKLAGLDINKTGGITLGMDELYKMHHHHHHVSKGEEENMASLPATHELHIFGSINGIDFDMVGQGTGNPNDGY :  80
NEMOb  : MLQNELALKLAGLDINKTGGGS---------HHHHHHVSKGEEENMASLPATHELHIFGSINGIDFDMVGQGTGNPNDGY :  71
NEMOf  : MLQNELALKLAGLDINKTGGGS---------HHHHHHVSKGEEENMASLPATHELHIFGSINGIDFDMVGQGTGNPNDGY :  71

                 *         *         *         *         *         *         *         *
                 |         |         |         |         |         |         |         |
NCaMP7 : EELNLKSTMGDLQFSPWILVPHIGYGFHQYLPYPDGMSPFQAAMVDGSGYQVHRTMQFEDGASLTVNYRYTYEGSHIKGE : 124
NEMOc  : EELNLKSTMGDLQFSPWILVPHIGYGFHQYLPYPDGMSPFQAAMVDGSGYQVHRTMQFEDGASLTVNYRYTYEGSHIKGE : 130
NEMOm  : EELNLKSTMGDLQFSPWILVPHIGYGFHQYLPYPDGMSPFQAAMVDGSGYQVHRTMQFEDGASLTVNYRYTYEGSHIKGE : 151
NEMOs  : EELNLKSTMGDLQFSPWILVPHIGYGFHQYLPYPDGMSPFQAAMVDGSGYQVHRTMQFEDGASLTVNYRYTYEGSHIKGE : 160
NEMOb  : EELNLKSTMGDLQFSPWILVPHIGYGFHQYLPYPDGMSPFQAAMVDGSGYQVHRTMQFEDGASLTVNYRYTYEGSHIKGE : 151
NEMOf  : EELNLKSTMGDLQFSPWILVPHIGYGFHQYLPYPDGMSPFQAAMVDGSGYQVHRTMQFEDGASLTVNYRYTYEGSHIKGE : 151

                 *         *         *         *         *         *         *         *
                 |         |         |         |         |         |         |         |
NCaMP7 : AQVEGTGFPADGPVMTNSLTAEAHDQLTEEQIAEFKEAFSLFDKDGDGTITTKELGTVMRSLGQNPTEAELRVMIIEVDA : 204
NEMOc  : AQVEGTGFPADGPVMTNSLTAEAHDQLTEEQIAEFKEAFSLFDKDGDGTITTKELGTVMRSLGQNPTEAELRVMIIEVDA : 210
NEMOm  : AQVEGTGFPADGPVMTNSLTAEAHDQLTEEQIAEFKEAFSLFDKDGDGTITTKELGTVMRSLGQNPTEAELRVMIIEVDA : 231
NEMOs  : AQVEGTGFPADGPVMTNSLTAEAHDQLTEEQIAEFKEAFSLFDKDGDGTITTKELGTVMRSLGQNPTEAELRVMIIEVDA : 240
NEMOb  : AQVEGTGFPADGPVMTNSLTAEAHDQLTEEQIAEFKEAFSLFDKDGDGTITTKELGTVMRSLGQNPTEAELRVMIIEVDA : 231
NEMOf  : AQVEGTGFPADGPVMTNSLTAEAHDDLTEEQIAEFKEEFSLFDKDGDGTITTKELGTVFRSLGQNPTEAELRVMIIEVDA : 231

                 *         *         *         *         *         *         *         *
                 |         |         |         |         |         |         |         |
NCaMP7 : DGDGTLDFPEFLAMMARKMKYRDTEEEIREAFGVFDKDGNGYIGAAELRHVMTNLGEKLTDEEVGELIREADIDGDGQVN : 284
NEMOc  : DGDGTLDFPEFLAMMARKMKYRDTEEEIREAFGVFDKDGNGYIGAAELRHVMTNLGEKLTDEEVGELIREADIDGDGQVN : 290
NEMOm  : DGDGTLDFPEFLAMMARKMKYRDTEEEIREAFGVFDKDGNGYIGAAELRHVMTNLGEKLTDEEVGELIREADIDGDGQVN : 311
NEMOs  : DGDGTLDFPEFLAMMARKMKYRDTEEEIREAFGVFDKDGNGYIGAAELRHVMTNLGEKLTDEEVGELIREADIDGDGQVN : 320
NEMOb  : DGDGTLDFPEFLAMMARKMKYRDTEEEIREAFGVFDKDGNGYIGAAELRHVMTNLGEKLTDEEVGELIREADIDGDGQVN : 311
NEMOf  : DGDGTLDFPEFLAMMARKMKYRDTEEEIREAFGVFDKDGNGYIGAAELRHVMTNLGEKLTDEEVGELIREADIDGDGQVN : 311

                 *         *         *         *         *         *         *         *
                 |         |         |         |         |         |         |         |
NCaMP7 : YEEFVQMMTAKGGSGGGSSSRRKWNKAGHAVRAIGRLSSMYFADWCVSKKTCPNDKTIVSTFKWAFITDNGKRYRSTART : 364
NEMOc  : YEEFVQMMTAKGGGGS--RRKWNKAGHAVRAIGRLSSTYFADWCVSKKTCPNDKTIVSTFKWAFITDNGKRYRSTART : 366
NEMOm  : YEEFVQMMTAKGGSGGG---RRKWNKAGHAVRAIGRLSSTYFADWCVSKKTCPNDKTIVSTFKWAFITDNGKRYRSTART : 388
NEMOs  : YEEFVQMMTAKGGSGGGSSSRRKWNKAGHAVRAIGRLSSTYFADWCVSKKTCPNDKTIVSTFKWAFITDNGKRYRSTART : 400
NEMOb  : YEEFVQMMTAKGGSGGGSSSRRKWNKAGHAVRAIGRLSSTYFADWCVSKKTCPNDKTIVSTFKWAFITDNGKRYRSTART : 391
NEMOf  : YEEFVQMMTAKGGGGS---RRKWNKAGHAVRAIGRLSSTYFADWCVSKKTCPNDKTIVSTFKWAFITDNGKRYRSTART : 387

                 *         *         *         *         *
                 |         |         |         |         |
NCaMP7 : TYTFAKPMAANYLKNQPMYVFRKTELKHSKTELNFKEWQKAFTDVMGMDELYK : 417
NEMOc  : TYTFAKPMAANYLKNQPMYVFRKTELKHSKTELNFKEWQKAFTDVMGMDELYK : 419
NEMOm  : TYTFAKPMAANYLKNQPMYVFRKTELKHSKTELNFKEWQKAFTDVMGMDELYK : 441
NEMOs  : TYTFAKPMAANYLKNQPMYVFRKTELKHSKTELNFKEWQKAFTDVMGMDELYK : 453
NEMOb  : TYTFAKPMAANYLKNQPMYVFRKTELKHSKTELNFKEWQKAFTDVMGMDELYK : 444
NEMOf  : TYTFAKPMAANYLKNQPMYVFRKTELKHSKTELNFKEWQKAFTDVMGMDELYK : 440
```

**Extended Data Fig. 1 | Alignment of the primary sequences of the indicated NCaMP7 and NEMO variants.** Mutations in NEMO related to NCaMP7 are highlighted in red. Changes of linker between CaM and M13 of NEMO related to NCaMP7 are shown in pink. The font colors correspond to the colors in the pattern diagram in Fig. 1a.

**A**

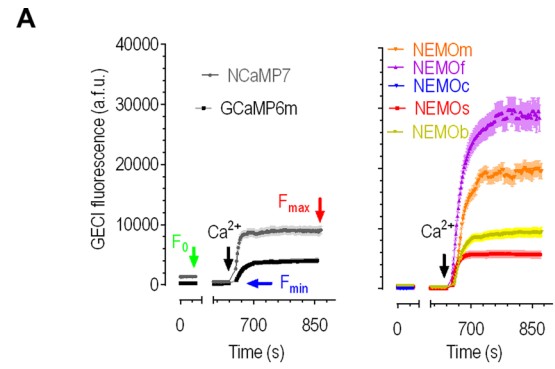

**B**

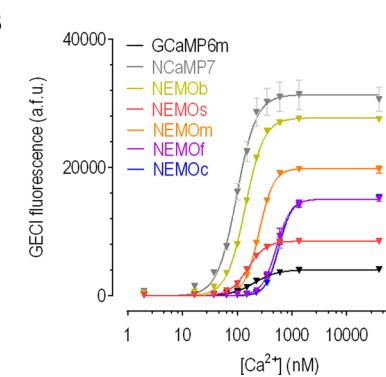

**Extended Data Fig. 2 | *In cellulo and in vitro* responses of NEMO sensors.** (**a**) Typical raw fluorescence readings obtained during the screening of NEMO indicators, corresponding to traces shown in Fig. 1c. GCaMP6m and NCaMP7 (left), or selected NEMO variants (right). (**b**) Representative raw traces of *in vitro* dose–response curves of GECIs, corresponding to those shown in Fig. 1e. (n = 3 independent biological replicates; >17 cells per repeat). Data shown as mean ± s.e.m.

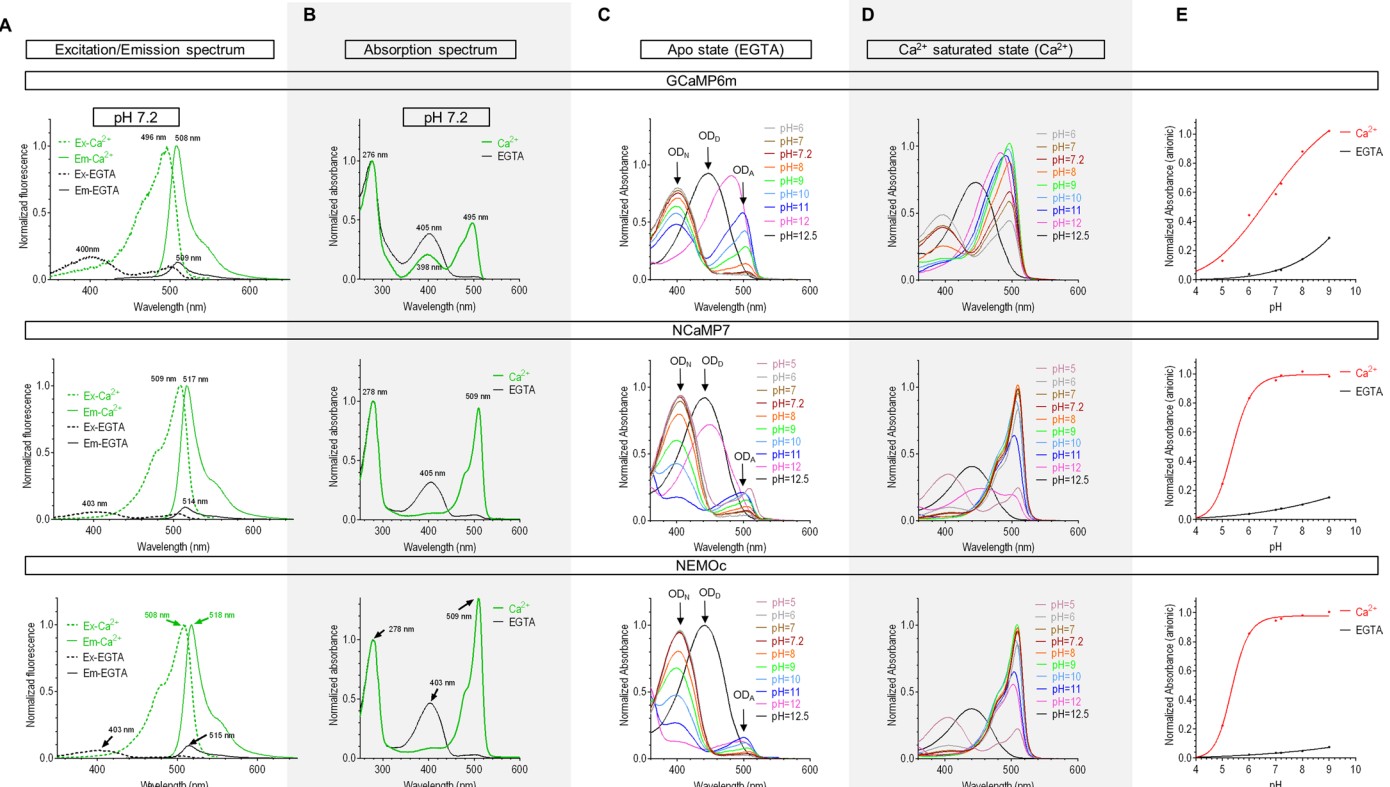

**Extended Data Fig. 3 | Spectral properties of NEMOc, NCaMP7 and GCaMP6m.** (**a**, **b**) At pH7.2, typical traces of excitation spectrum, emission spectrum (A), and absorption spectrum (B). (**c**, **d**) pH-dependence of absorption spectrum at the apo state (C) and Ca²⁺ saturated conditions (D). (**e**) Statistics for data shown in panels C&D. n = 3 independent biological replicates. The corresponding absorbance (OD) in protonated ($OD_N$), deprotonated ($OD_A$), and denatured ($OD_D$) states were indicated by arrows. n = 3 independent replicates.

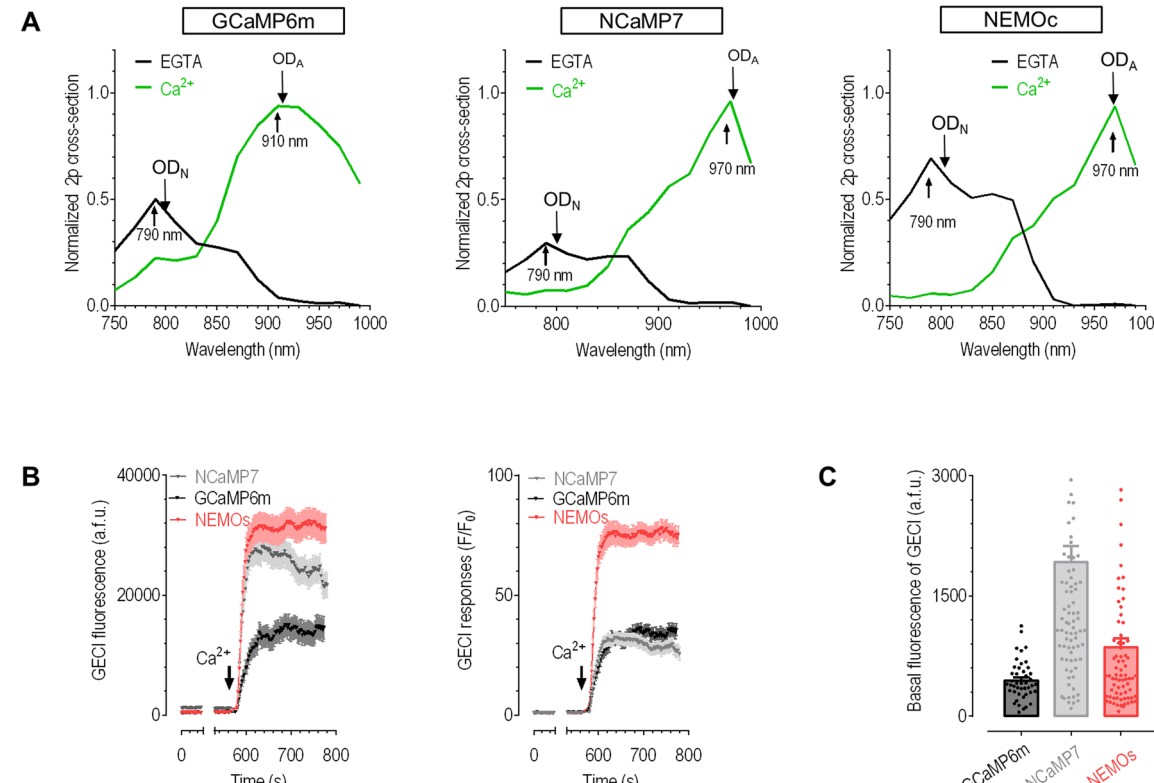

**Extended Data Fig. 4 | Two-photon spectral properties and performance of NCaMP7, GCaMP6m and NEMO sensors. (a)** Typical traces of normalized two-photon action cross-sections at pH 7.2. **(b)** Dynamic ranges of the indicated GECIs when expressed in HEK293 cells. **(c)** Statistics of basal fluorescence shown in panel (B). GCaMP6m, n = 45 cells; NEMOs, n = 76 cells; NCaMP7, n = 89 cells examined over 3 independent biological repeats. Data in (B) and (C) were shown as mean ±s.e.m.

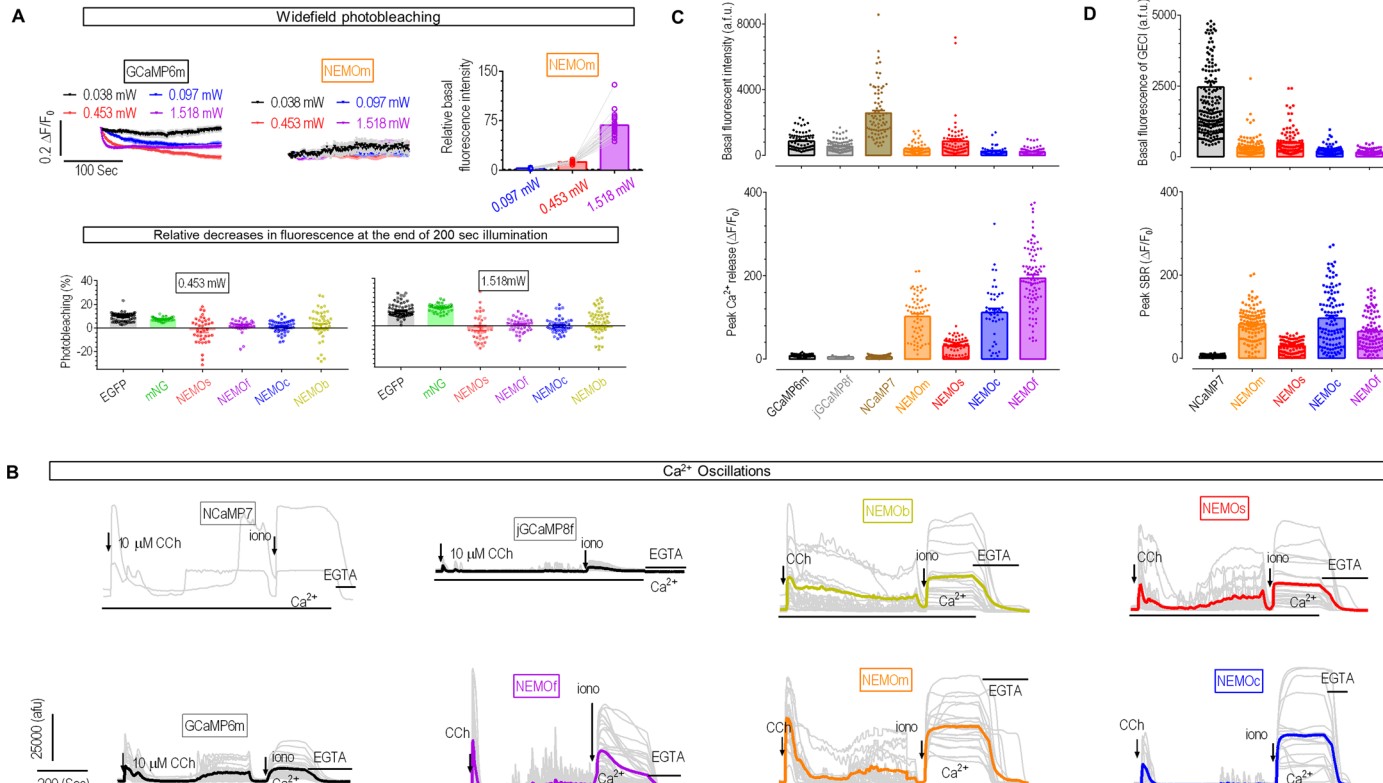

**Extended Data Fig. 5 | NEMO performance in HEK293 cells. (a)** Widefield photobleaching curves. GFP excitation light (470 ± 11 nm) was used. Top, two panels on the left, GCaMP6m and NEMOm signals; right, basal fluorescence intensity relative to those excited with 0.038 mW light; Bottom panels, statistics showing the relative reduction in fluorescence at the end of 200 sec illumination with dim (left) or strong (right) light. Top right (n = 27 cells); Bottom left (mNG, n = 42 cells; EGFP, n = 76 cells; NEMOs, n = 38 cells; NEMOf, n = 37 cells; NEMOc, n = 50 cells; NEMOb, n = 47 cells); Bottom right (mNG, n = 46 cells; EGFP, n = 72 cells; NEMOs, n = 35 cells; NEMOf, n = 42 cells; NEMOc, n = 47 cells; NEMOb, n = 56 cells). **(b)** Typical fluorescence oscillations of GECI variants induced by

CCh (10 μM). Recordings from the same cells (as presented in Fig. 2a) were shown. Grey lines, responses of individual cells; thick lines, mean responses. **(c)** Statistics of results shown in panel B and Fig. 2a. Left, the basal fluorescence; right, peak of the first $Ca^{2+}$ release. (GCaMP6m, n = 73 cells; NEMOs, n = 71 cells; NEMOm, n = 63 cells; NEMOc, n = 46 cells; NCaMP7, n = 77 cells; jGCaMP8f, n = 93 cells; NEMOf, n = 93 cells). **(d)** Statistics of results shown in Fig. 2b. Left, mean basal fluorescence; right, the peak SOCE response. n = 3 independent biological replicates. (NCaMP7, n = 192 cells; NEMOc, n = 110 cells; NEMOm, n = 117 cells; NEMOf, n = 98 cells; NEMOs, n = 99 cells). Data from (A), (C) and (D) were from 3 independent biological replicates, and shown as mean ±s.e.m.

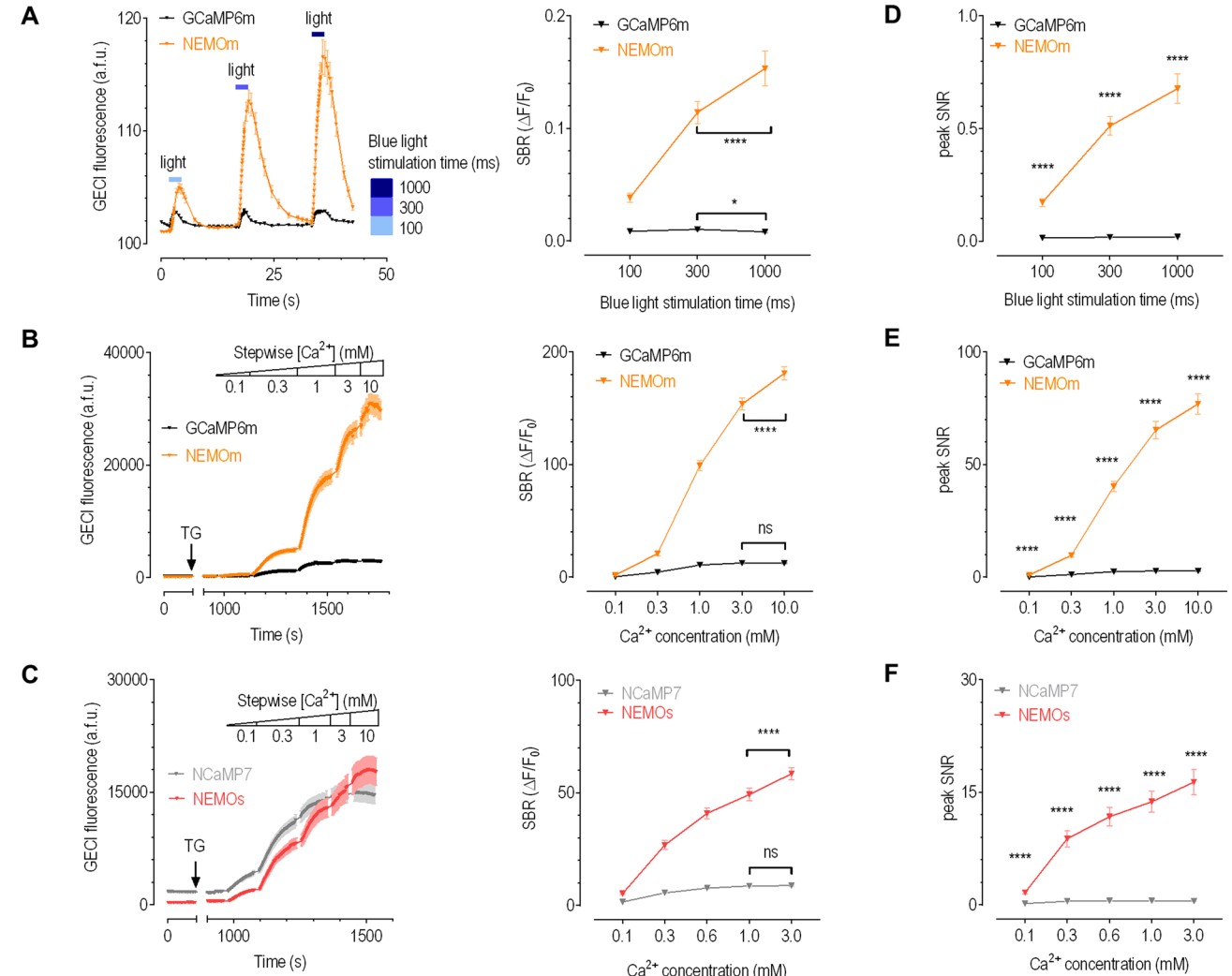

**Extended Data Fig. 6 | NEMO performance in resolving Ca²⁺ signals that span a wide range of magnitude.** (**a**) Dim blue light stimulation induced GECI responses in HeLa cells co-expressing Opto-CRAC and GCaMP6m or NEMOs. Left, typical traces; right, statistics (NEMOm, 1000 ms vs 300 ms, ****, p = 4.3E-7; GCaMP6m, 1000 ms vs 300 ms, p = 0.0178). To avoid direct activation with 488-nm light, minimal (1%) excitation input at 488 nm was used. Three 15-cycles of 2% 473-nm stimulation light with different exposure time were sequentially applied to photo-induce more Ca²⁺ influxes. 100 ms for the first one, 300 ms for the second, and 10000 ms for the third cycle of photo-stimulation. (**b**, **c**) SOCE indicated by GECIs in HEK293 cells bathed in stepwise external Ca²⁺. (B, GCaMP6m vs NEMOm; C, NCaMP7 vs NEMOs). Left, typical traces;

right, statistics. (For B right panel, NEMOm, 10 mM vs 3 mM, ****, p = 9.28E-24; GCaMP6m, 10 mM vs 3 mM, p = 0.7917. For C right panel, NEMOs, 3 mM vs 1 mM, ****, p = 6.152E-12; NCaMP7, 3 mM vs 1 mM, p = 0.0681.) (**d-f**) Statistics of panels A-C showing SNR of GECIs. (For D panel, 100 ms, p = 2E-15; 300 ms, p = 1.03E-22; 1000 ms, p = 5.06E-18. For E panel, 0.1 mM, p = 6.95E-23; 0.3 mM, p = 7.01E-26; 1 mM, p = 3.79E-36; 3 mM, p = 1.56E-36; 10 mM, p = 5.85E-37. For F panel, 0.1 mM, p = 1.13E-13; 0.3 mM, p = 5.13E-16; 0.6 mM, p = 8.85E-20; 1 mM, p = 1.44E-21; 3 mM, p = 3.58E-22). (A-C) Paired Student's *t*-test, two-tailed. (D-F) Unpaired Student's *t*-test, two-tailed. n = 3 independent biological replicates. All data in this figure were shown as mean ± s.e.m.

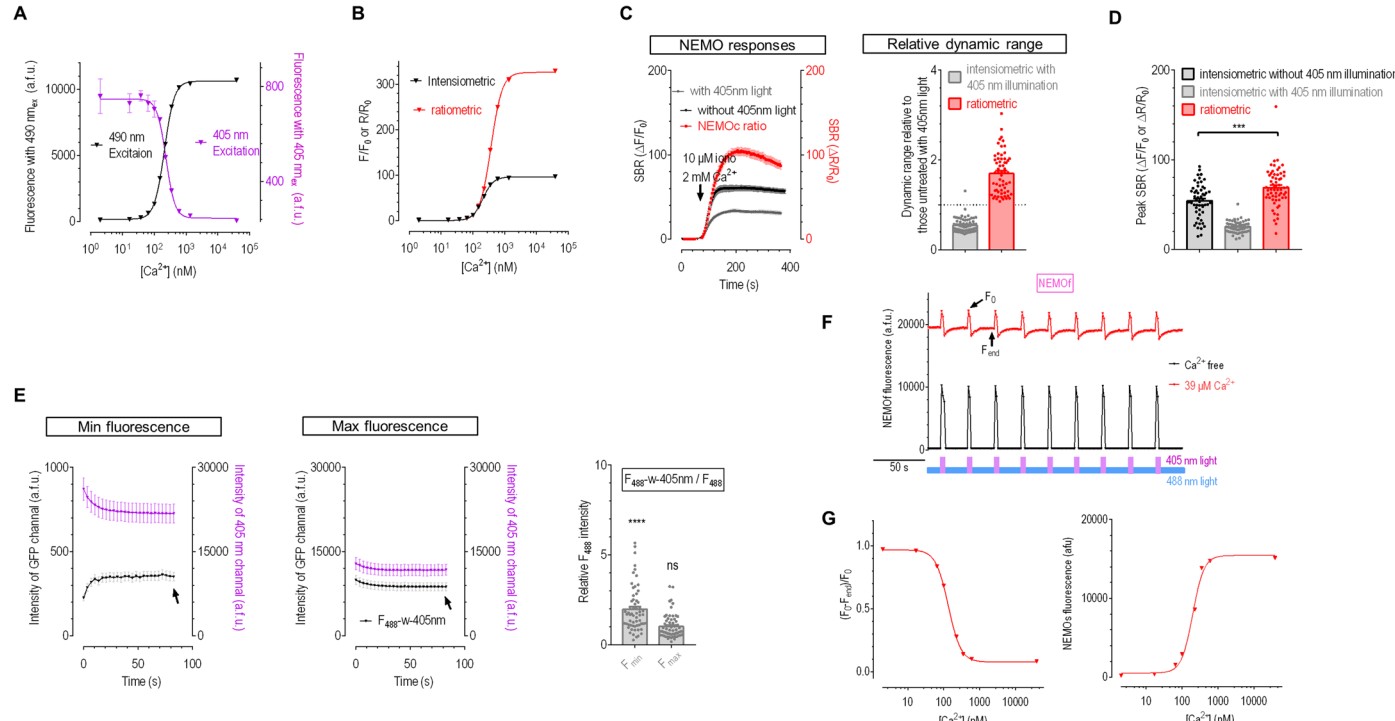

**Extended Data Fig. 7 | Ratiometric or photochromic responses of NEMO sensors.** (**a**, **b**) In vitro $Ca^{2+}$ titration curves. (A) 405 nm ($F_{405}$) or 490 nm ($F_{490}$) excited fluorescence responses. n = 3 independent measurements. Error bars correspond to s.e.m. (B) Comparison of instensiometric or ratiometric (R = ($F_{490}/F_{405}$) responses of NEMOs. n = 3 independent measurements. (**c–e**) Evaluation of ratiometric responses of NEMOs expressed in HEK cells. 3 independent biological repeats. (C) Maximal NEMOs responses with or without 405 nm illumination. Left, typical traces; right, statistics. (D) Statistics of results in Fig. 2d. (***, p = 0.0002, unpaired Student's t-test, two-tailed) (Intensiometric without 405 nm illumination, n = 54 cells; intensiometric with 405 nm illumination and ratiometric, n = 61 cells). (E) Effect of 405 nm light illumination on $F_{405}$ or $F_{488}$. Left, representative NEMOs responses when cytosolic $Ca^{2+}$ were mostly exhausted by 10 min bath in imaging solution containing 10 μM ionomycin and 1 mM EGTA. Middle, typical traces showing NEMOc responses bathed in 10 μM ionomycin and 2 mM $Ca^{2+}$. Right, statistics. $F_{488}$-w-405nm,

NEMOs signal excited with 488 nm laser that were exposed to 405 nm ($F_{min\,405}$ vs $F_{488}$ and $F_{max\,405}$ vs $F_{488}$, ****, p = 4.32E-6 and p = 0.88, unpaired Student's t-test, two-tailed; n = 61 cells). (C, E; n = 61 cells) (**f, g**) In vitro $Ca^{2+}$ titration of NEMOf with intermittent photochromism-enabled absolute quantification (iPEAQ) method. n = 3 independent measurements. (F) Typical traces showing the responses of 488nm-excited NEMOf to repeated illumination by 405 nm light. $F_0$ and $F_{end}$, indicated by arrows, peak and minimum fluorescence intensities used to calculate values of the photochromism contrast $(F_0 - F_{end}) / F_{end}$, or $(\Delta F / F_0)_{hv}$. (G) $Ca^{2+}$ titration curves. Left panel, the photochromism contrast $((\Delta F / F_0)_{hv}$ - $Ca^{2+}$ titration curve that enabled conversion of measured basal photochromism contrast to basal $Ca^{2+}$ concentration of a NEMOs-expressing cell; Right panel, fluorescence - $Ca^{2+}$ titration curve that enabled the subsequent determination of changes in absolute $Ca^{2+}$ concentrations with the calculated basal $Ca^{2+}$ level and recorded NEMOs fluorescence response curve. Except for (F), all other panels in this figure were shown as mean ± s.e.m.

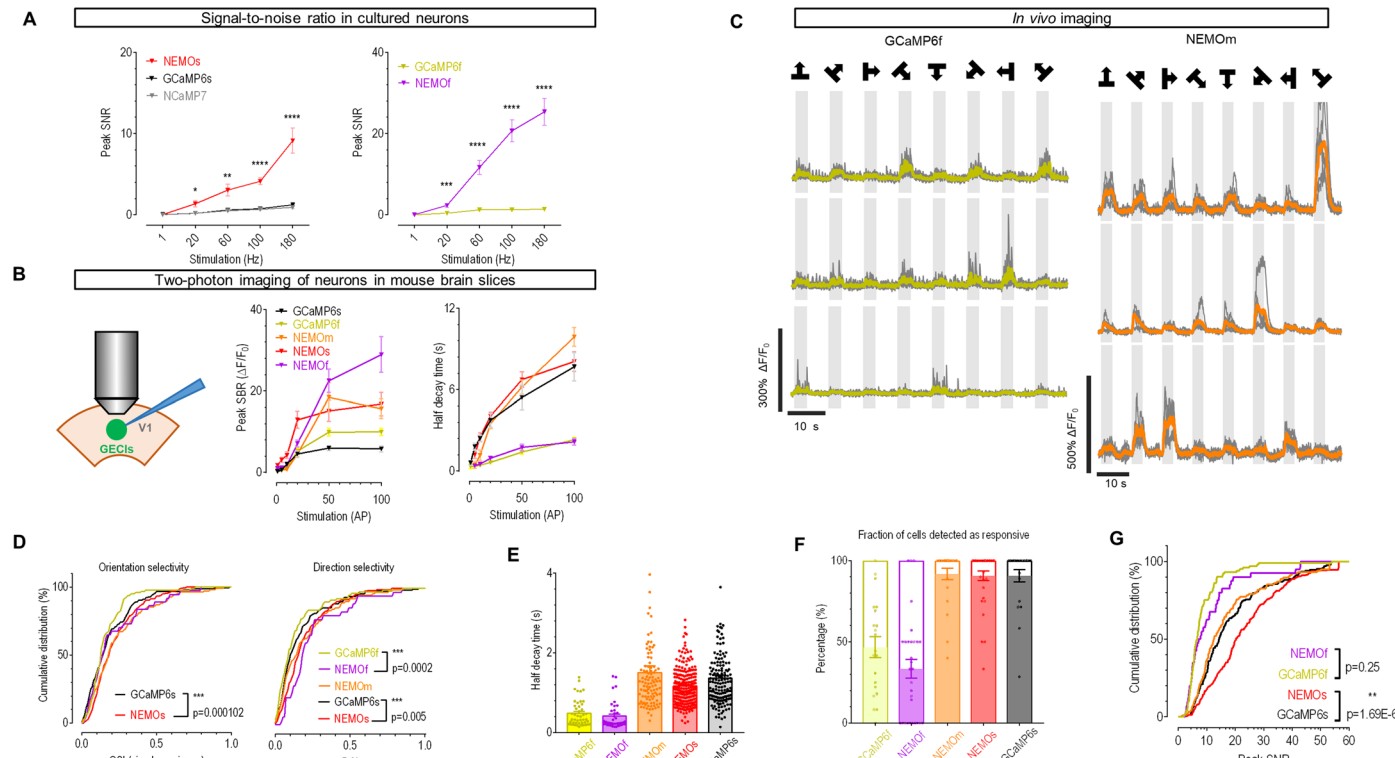

**Extended Data Fig. 8 | Performance of NEMO sensors in rodent neurons.**
(**a**) Signal-to-noise ratio (SNR) of GECIs in cultured neurons (unpaired Student's *t*-test, two-tailed; Left panel, NEMOs vs NCaMP7 at 20 Hz, *, p = 0.0182; 60 Hz, **, p = 0.0035; 100 Hz, ****, p = 2.98E-8; 180 Hz, ****, p = 1.50E-6; right panel, NEMOf vs GCaMP6f at 20 Hz, ***, p = 1.05E-4; 60 Hz, ****, p = 2.24E-7, 100 Hz, ****, p = 2.41E-9; 180 Hz, ****, p = 1.24E-6) (For stimulation at varied frequencies, NCaMP7, n = 12, 12, 12, 14, 12 cells. The 'n' values of other sensors were equal to Fig. 3. Each GECI measurement set was analyzed from multiple dendrites of neurons in three different primary hippocampal neuron cultures. (**b**) Fluorescence responses of NEMO variants in neurons of acute mouse brain slices. Left, cartoon showing the set up of two-photon fluorescence imaging under a whole-cell patch clamp configuration. Middle, statistics showing the peak SBR-frequency of action potentials (AP) (NEMOf vs GCaMP6s at 50 Hz, p = 3.5E-4; at 100 Hz, p = 1.4E-5; unpaired Student's t-test, two-tailed) (GCaMP6f, n = 57 cells from 3 mice; GCaMP6s, n = 51 cells from 2 mice; NEMOs, n = 30 cells from 4 mice; NEMOm, n = 45 cells from 3 mice; NEMOf, n = 54 cells, from 3 mice); right, half decay time of NEMO variants under different intensities of stimuli (For stimulation from 5-100 AP, GCaMP6f, n = 12,14,18,18,18 cells from 3 mice; GCaMP6s, n = 17 cells

from 2 mice; NEMOm, n = 2,4,14,16,16 cells from 3 mice; NEMOs, n = 8,9,10,10,9 cells from 4 mice; NEMOf, n = 5,9,15,18,17 cells from 3 mice; for stimulation at 1 AP, GCaMP6s, n = 10 cells from 2 mice; GCaMP6f, n = 3 cells from 3 mice). (**c-f**) *In vivo* two-photon imaging of visual cortex neurons in response to drift gratings. (C) Typical responses; (D-F) Statistics of results shown in Fig. 4a–c; (D) Cumulative distribution of orientation (left, ****, p = 0.000102) or direction (right, for NEMOf vs GCaMP6f and NEMOs vs GCaMP6s, ****, p = 0.0002; ***, p = 0.005) selectivity (Kolmogorov–Smirnov test, two-tailed). (E) Statistics of half decay time (GCaMP6f, n = 46 cells from 2 mice; GCaMP6s, n = 157 cells from 3 mice; NEMOs, n = 223 cells from 4 mice; NEMOm, n = 105 cells from 3 mice; NEMOf, n = 40 cells from 3 mice). (F) Statistics showing fraction of responsive cells (NEMOf vs GCaMP6f, p = 0.1356; NEMOs vs GCaMP6s, p = 0.9805, unpaired Student's t-test, two-tailed) (GCaMP6f, n = 21 cells from 2 mice; GCaMP6s, n = 23 cells from 3 mice; NEMOs, n = 35 cells from 4 mice; NEMOm, n = 24 cells, from 3 mice; NEMOf, n = 30 cells from 3 mice). (**g**) Cumulative distribution of SNR (NEMOf vs GCaMP6f and NEMOs vs GCaMP6s, p = 0.25; ****, p = 1.69E-6, Kolmogorov-Smirnov test, two-tailed). At least n = 3 independent biological replicates. Data in A, B, E and F panels were shown as mean ± s.e.m.

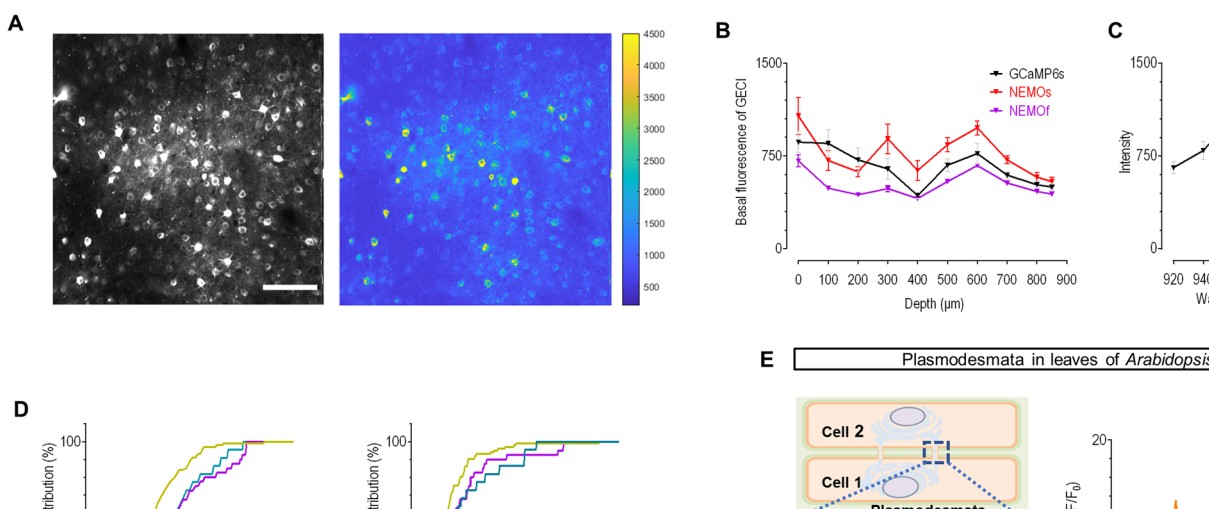

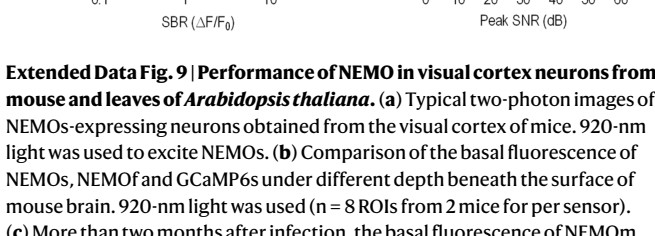

**Extended Data Fig. 9 | Performance of NEMO in visual cortex neurons from mouse and leaves of *Arabidopsis thaliana*.** (**a**) Typical two-photon images of NEMOs-expressing neurons obtained from the visual cortex of mice. 920-nm light was used to excite NEMOs. (**b**) Comparison of the basal fluorescence of NEMOs, NEMOf and GCaMP6s under different depth beneath the surface of mouse brain. 920-nm light was used (n = 8 ROIs from 2 mice for per sensor). (**c**) More than two months after infection, the basal fluorescence of NEMOm of the same set of cells excited by light at different wavelengths was quantified (n = 20 cells). Data in (B) and (C) were shown as mean ± s.e.m. (**d**) Cumulative

distribution of SBR (left) or SNR (right) of NEMOf excited at 920-nm (n = 40 cells) or 980 nm (n = 22 cells). GCaMP6f and NEMOf data were replotted from Extended Data Fig. 8g. (Left, NEMOf vs NEMOf-980, p = 0.95; Right, GCaMP6f vs NEMOf-980, \***, p = 0.0005; NEMOf vs NEMOf-980, \*\*, p = 0.008; Kolmogorov-Smirnov test, two-tailed). **e**) Ca$^{2+}$ oscillations near plasmodesmata in the leaves of *Arabidopsis thaliana* when excited at 488 nm. NEMOm was tagged with PDLP1, a plasmodesmata marker. Scale bar, 5 μm. Left, cartoon illustration showing the structure of plasmodesmata; right, typical traces. At least n = 3 independent biological replicates.

# natureportfolio

# Reporting Summary

## Statistics

For all statistical analyses, confirm that the following items are present in the figure legend, table legend, main text, or Methods section.

| n/a | Confirmed | |
|---|---|---|
| ☐ | ☒ | The exact sample size (*n*) for each experimental group/condition, given as a discrete number and unit of measurement |
| ☐ | ☒ | A statement on whether measurements were taken from distinct samples or whether the same sample was measured repeatedly |
| ☐ | ☒ | The statistical test(s) used AND whether they are one- or two-sided<br>*Only common tests should be described solely by name; describe more complex techniques in the Methods section.* |
| ☒ | ☐ | A description of all covariates tested |
| ☒ | ☐ | A description of any assumptions or corrections, such as tests of normality and adjustment for multiple comparisons |
| ☐ | ☒ | A full description of the statistical parameters including central tendency (e.g. means) or other basic estimates (e.g. regression coefficient) AND variation (e.g. standard deviation) or associated estimates of uncertainty (e.g. confidence intervals) |
| ☐ | ☒ | For null hypothesis testing, the test statistic (e.g. *F*, *t*, *r*) with confidence intervals, effect sizes, degrees of freedom and *P* value noted<br>*Give P values as exact values whenever suitable.* |
| ☒ | ☐ | For Bayesian analysis, information on the choice of priors and Markov chain Monte Carlo settings |
| ☒ | ☐ | For hierarchical and complex designs, identification of the appropriate level for tests and full reporting of outcomes |
| ☒ | ☐ | Estimates of effect sizes (e.g. Cohen's *d*, Pearson's *r*), indicating how they were calculated |

*Our web collection on statistics for biologists contains articles on many of the points above.*

## Software and code

Policy information about availability of computer code

| Data collection | Data of confocal imaging and iPEAQ measurements were collected with Zen 2.1 software. All time-lapse fluorescence imaging were carried out using SlideBook 6.0.23. Results of biophysical characterization were collected by UVProbe, Fluoracle and SoftMax Pro v7.x. Data of Ca2+ imaging in hippocampal neurons were collected with NIS-Elements AR 5.10.00. Two-photon laser Ca2+ imaging were performed with scanbox 4.1 or FV10-ASW 4.2, and electrophysiology data were collected by pClampex 10.3. Visual stimulus was controlled by LabVIEW 8.5. Optical fiber recording was achieved with ORFS V2_14397. |
|---|---|
| Data analysis | The acquired images or data were analyzed by MATLAB 2014a or OLYMPUS FV10-ASW and plotted with Prism7 software. We mostly used customized Matlab scripts for very basic matrix operations, such as extracting data points from excel spread sheets that contain raw data, calculations, and commonly used statistical analysis. The related codes can be shared upon reasonable requests. |

For manuscripts utilizing custom algorithms or software that are central to the research but not yet described in published literature, software must be made available to editors and reviewers. We strongly encourage code deposition in a community repository (e.g. GitHub). See the Nature Portfolio guidelines for submitting code & software for further information.

## Data

Policy information about availability of data

All manuscripts must include a data availability statement. This statement should provide the following information, where applicable:

- Accession codes, unique identifiers, or web links for publicly available datasets
- A description of any restrictions on data availability
- For clinical datasets or third party data, please ensure that the statement adheres to our policy

All data generated or analyzed during this study are included in this published article (and its supplementary information files). Source Data are available with this publication. Key NEMO plasmids are available via Addgene (189930 ~ 189934). The coding sequence of NEMO sensors have been deposited to GenBank (NEMOf, OQ626715; NEMOc, OQ626716; NEMOb, OQ626717; NEMOm, OQ626718; NEMOs, OQ626719).

## Human research participants

Policy information about studies involving human research participants and Sex and Gender in Research.

| Reporting on sex and gender | N/A |
| --- | --- |
| Population characteristics | N/A |
| Recruitment | N/A |
| Ethics oversight | N/A |

Note that full information on the approval of the study protocol must also be provided in the manuscript.

# Field-specific reporting

Please select the one below that is the best fit for your research. If you are not sure, read the appropriate sections before making your selection.

☒ Life sciences ☐ Behavioural & social sciences ☐ Ecological, evolutionary & environmental sciences

For a reference copy of the document with all sections, see nature.com/documents/nr-reporting-summary-flat.pdf

# Life sciences study design

All studies must disclose on these points even when the disclosure is negative.

| Sample size | We determined sample size based on well established studies in the field and the animal-to-animal variability observed during the experiments. The sample size in detecting the performance of transiently expressed NEMO sensors in non-excitable HEK-293 cells was at least 3 times repeats with 9~20 cells per repeat. The sample size in detecting the responses of NEMO variants in dissociated rat hippocampus neurons excited by electric field stimulation was at least 10 neurons in three different primary hippocampal neuron cultures. The sample size of tail-pinching stimulus and optical fiber recording was 97~101 cells from 3 mice. The sample size of simultaneous two-photon laser $Ca^{2+}$ imaging and electrophysiology in visual cortical slices was 40~223 cells from at least 3 mice. Corresponding literatures provided in methods. |
| --- | --- |
| Data exclusions | No data were excluded from the analyses. |
| Replication | At least three labs have independently confirmed the superior SBR of NEMO sensors. Experiments have been done with several mice or cells to check the reproducibility of our results. At least three independent repeats were carried out for each set of experiments. |
| Randomization | Allocation of samples was random. |
| Blinding | The investigators were blinded to group allocation during data collection and analysis. |

# Reporting for specific materials, systems and methods

We require information from authors about some types of materials, experimental systems and methods used in many studies. Here, indicate whether each material, system or method listed is relevant to your study. If you are not sure if a list item applies to your research, read the appropriate section before selecting a response.

## Materials & experimental systems

| n/a | Involved in the study |
|-----|----------------------|
| ☒ ☐ | Antibodies |
| ☐ ☒ | Eukaryotic cell lines |
| ☒ ☐ | Palaeontology and archaeology |
| ☐ ☒ | Animals and other organisms |
| ☒ ☐ | Clinical data |
| ☒ ☐ | Dual use research of concern |

## Methods

| n/a | Involved in the study |
|-----|----------------------|
| ☒ ☐ | ChIP-seq |
| ☒ ☐ | Flow cytometry |
| ☒ ☐ | MRI-based neuroimaging |

## Eukaryotic cell lines

Policy information about cell lines and Sex and Gender in Research

| | |
|---|---|
| Cell line source(s) | Hek 293 cells used in this experiment were derived from ATCC (American Type Culture Collection, cat#: crl-1573). HeLa cells also from ATCC (cat#: CL 0101). |
| Authentication | Authentication was guaranteed by the provider. We verified the cell lines based on morphology. |
| Mycoplasma contamination | The cell lines were not tested for mycoplasma contamination. |
| Commonly misidentified lines (See ICLAC register) | The study did not involve commonly misidentified cell lines. |

## Animals and other research organisms

Policy information about studies involving animals; ARRIVE guidelines recommended for reporting animal research, and Sex and Gender in Research

| | |
|---|---|
| Laboratory animals | E18 Wistar rats of either gender, C57BL/6 mice at postnatal 15-20 days (P15-P20) and adult mice(> P50) of either gender and male C57BL/6 mice (7 weeks old, weighing 20-25 g). C57BL/6 mice were housed in a 12 h light/dark cycle. Food and water were provided ad libitum. The temperature of the room was controlled at 20–25 °C, and the humidity was maintained at 45–60%. |
| Wild animals | The study did not involve wild animals. |
| Reporting on sex | Male or female mice/rats were used randomly |
| Field-collected samples | The studies did not involve samples collected in field. |
| Ethics oversight | Animal experiments were approved by the Animal Experimental Ethics Committee of Beijing Normal University and University of Science and Technology of China. |

Note that full information on the approval of the study protocol must also be provided in the manuscript.

