## [Peer Review File · Nature Methods]

Peer Review Information

Manuscript Title: Engineering of NEMO as calcium indicators with large dynamics and high sensitivity

Corresponding author name(s): Yubin Zhou, Ai-Hui Tang, Xiaohui Zhang, Youjun Wang

Reviewer Comments & Decisions:

Decision Letter, initial version:
--

13th Sep 2022

Dear Professor Wang,

Thank you for your patience during the review process. I apologize for the delays. We were waiting to hear back from an additional reviewer, but this reviewer did not provide their input. Your Article, "Engineering of NEMO as calcium indicators with ultra-high dynamics and sensitivity", has now been seen by two reviewers. As you will see from their comments below, although the reviewers find your work of considerable potential interest, they have raised a number of concerns. We are interested in the possibility of publishing your paper in Nature Methods, but would like to consider your response to these concerns before we reach a final decision on publication. We therefore invite you to revise your manuscript to address these concerns.

- * include a point-by-point response to the reviewers and to any editorial suggestions
- * please underline/highlight any additions to the text or areas with other significant changes to facilitate review of the revised manuscript
- * address the points listed described below to conform to our open science requirements
- * ensure it complies with our general format requirements as set out in our guide to authors at

www.nature.com/naturemethods

* resubmit all the necessary files electronically by using the link below to access your home page

[REDACTED]

We hope to receive your revised paper within 6-8 weeks. If you cannot send it within this time, please let us know. In this event, we will still be happy to reconsider your paper at a later date so long as nothing similar has been accepted for publication at Nature Methods or published elsewhere.

OPEN SCIENCE REQUIREMENTS

REPORTING SUMMARY AND EDITORIAL POLICY CHECKLISTS

DATA AVAILABILITY

We strongly encourage you to deposit all new data associated with the paper in a persistent repository where they can be freely and enduringly accessed. We recommend submitting the data to discipline-

specific and community-recognized repositories; a list of repositories is provided here:
<http://www.nature.com/sdata/policies/repositories>

All novel DNA and RNA sequencing data, protein sequences, genetic polymorphisms, linked genotype and phenotype data, gene expression data, macromolecular structures, and proteomics data must be deposited in a publicly accessible database, and accession codes and associated hyperlinks must be provided in the "Data Availability" section.

MATERIALS AVAILABILITY

ORCID

Nature Methods is committed to improving transparency in authorship. As part of our efforts in this direction, we are now requesting that all authors identified as 'corresponding author' on published

papers create and link their Open Researcher and Contributor Identifier (ORCID) with their account on the Manuscript Tracking System (MTS), prior to acceptance. This applies to primary research papers only. ORCID helps the scientific community achieve unambiguous attribution of all scholarly contributions. You can create and link your ORCID from the home page of the MTS by clicking on 'Modify my Springer Nature account'. For more information please visit www.springernature.com/orcid.

Best regards,
Nina

Nina Vogt, PhD
Senior Editor
Nature Methods

Reviewers' Comments:

Reviewer #1:

Remarks to the Author:

In this manuscript, Wang et.al., developed a series of mNeonGreen-based Ca²⁺ indicators having increased dynamic range with several accessory functions including high brightness and fast kinetics. Properties of NEMO are validated by well-designed in-vitro and in-cell analysis, demonstrating the high detectability of Ca²⁺ dynamics with improved SBR than those of existing GECIs. The authors also performed in-vivo analysis including 2-P microscopy and fiber photometry, clearly showing the potential utility of NEMO in living animals. Overall the manuscript is well-organized to emphasize the advantages of NEMO by focusing on the expanded dynamic range. However, the manuscript still has room for improvement in some data presentation, in which the lack of analysis details made reviewers hard to validate the potentially excellent work, and part of the logic is not simple and logical to draw a clear conclusion as pointed below.

Comments:

1) Authors demonstrated the advantages of NEMO by primarily focusing on their increased DR which allowed the detection of Ca²⁺ dynamics with a larger signal change than existing indicators. In principle, the merits of expanded dynamic range should include a resolvability of similarly large signals which can be miss-evaluated to have the identical magnitude by indicators with a small dynamic range. Can authors strengthen this point?

2) When compared to parental NCaMP7, in vitro characterization (Table S4) showed that the increased dynamic range of NEMO was achieved by the decrease of min-brightness, not by an increase of max-brightness. Generally, low brightness causes the noisy baseline intensity that would spoil the signal-to-noise-ratio (SNR). Thus, a comparative analysis of SNR is essential to convince the utility of NEMO

series in in-cell and in-vivo demonstrations. If NEMO shows better SNR than NCaMP7, it could be a strong point of this work to be emphasized.

3) Authors could do a more straightforward presentation for NEMO's properties. It is notoriously confusing to round about the data between "in cell" and "in vitro" analysis for many variants. Concerning this issue, increased max-brightness of mNG-based indicator than GCaMP would be caused by the complete shifting of chromophore state to the bright anionic form. But this is not a unique feature of NEMO since NCaMP7 and related mNG-GECO already possessed this feature. Consequently, the uniqueness of NEMO might become the reduced brightness at low Ca²⁺ condition (and faster dissociation kinetics). To help readers' understanding of this point, data measured by the brightness would be needed especially for Fig1B and 1E in addition to F/F₀. Fig1B and 1D are not convincing in the absence of the correction of the expression level of given indicators among cells and constructs. I also wonder why NEMO shows larger DR in cells than in vitro?

4) For in-vivo analysis, NEMO would have increased detectability of Ca²⁺ change than GCaMP6, but it is not convincing in the absence of comparative analysis of SNR. Also, readers may wonder whether NEMO shows higher performance than NCaMP7 and related mNG-GECO in detecting neuronal activities with expecting increased SBR, SNR and kinetics (and K_d). These could be tested at least for cultured neurons and hopefully for brain slices. Additionally, the optical properties of NEMO protein at two-photon excitation are informative.

5) Application of NEMO in Fig2E is potentially excellent, but it is hardly evaluable in the absence of experimental details. From the viewpoint of K_d (528nM, in vitro), I also wonder about the reliability of the maximum or minimum concentration of Ca²⁺ in a cell. Provide detailed analysis schemes and discussion including in-cell calibration of Ca²⁺ affinity to ensure readers' understanding how iPEAQ using NEMO reliably determines Ca²⁺ concentration for 4μM at the peak and below 100nM at resting.

6) Technical comments on Ca²⁺ titration: Are there any reasons to utilize HEPES? Experts in this field routinely utilize photostable MOPS rather than photo-unstable HEPS, since the latter is producible with H₂O₂ upon light illumination that can directly affect the property of indicators and indirectly affect the free Ca²⁺ concentration through the reduced buffering ability.

7) Please provide precise descriptions to avoid any ambiguities;

(a) L127: What is the clear answer for "Therefore, the larger dynamic range of NEMO indicators is not a mere reflection of lower basal fluorescence." ? (see also comment (c))

(b) L141-: specify the reason for comparing the molecular brightness in the presence of Mg²⁺.

(c) L147: "superior dynamic range of NEMO indicators can be explained by Ca²⁺-induced changes in both the proportion and the absolute brightness of the anionic fluorophores of NEMO variants." This statement is true when compared to GCaMP6, but is false when compared to NCaMP7.

(d) L155: specify the basis for 40x (FigS3 left) and 60x (FigS3 right) by referring appropriate Supplementary Figs, respectively.

(e) L158: specify the scenario requiring "strong illumination".

(f) L207: "directly indicating basal Ca²⁺ level" is not logical.

(g) L158: It is overstating that "collectively~ NEMO variants as highly photo-stable sensors", in the absence of comparative photobleaching analysis at saturating Ca²⁺ conditions.

(h) correct typos in SI;

-Data analysis section: the peak signal to ratio (SNR): "noise" is missing

-Legend for Supplemental Figure: Fig4. E) Left----Left.... "Right" is missing

Reviewer #2:

None

Reviewer #3:

Remarks to the Author:

Jia Li and colleagues report a new suite of GECIs called NEMO, or mNeonGreen-based calcium indicators, which are derived from NCaMP7 by modifying linker sequences, introducing calmodulin mutations, or including a nuclear export signal. All of these new NEMO sensors are shown to display extreme enhancements in dynamic range, both in vitro and in 293 cells, versus current sensors (e.g., GcaMP6, jGCaMP8), with several hundred-fold changes in fluorescence intensity between apo and Ca²⁺-bound states. The authors show that NEMO very robustly detects receptor-stimulated Ca²⁺ elevations, including GPCR-mediated oscillations. In cultured neurons, NEMO compares favorably with current GECIs in detecting single action potentials and dwarfs current best-in-class GECIs in response to increasing stimulation frequencies. Finally, the authors use NEMO to robustly visualize Ca²⁺ responses in the brains of awake mice via 2p imaging in the V1 visual cortex and fiber photometry in the corpus striatum. The author's work is thorough, with careful characterization experiments and benchmarking against relevant sensors that are currently in wide use. NEMO is undoubtedly an impressive new tool that will be of great interest and utility.

Specific comments:

- 1) In Figures 4 and S5, the authors show two-photon imaging data acquired from acute brain slices and the brains of awake mice to claim improved performance of their NEMO sensors for in vivo imaging. The authors are strongly encouraged to add in vitro 2p characterization of their sensors, similar to 1p characterization in Figures 1 and S2 to enable more robust and absolute evaluation of 2p performance.
- 2) Lines 135-149: While the overall conclusions drawn by the authors seem consistent, it is hard to follow the experiments performed to differentiate the proportion vs absolute brightness. Neither this paragraph, nor Supplementary Figure 2, Supplementary Table 4 or the Material and Methods describe the experiments and data analysis in sufficient detail for the reader to follow. Please include a more precise description of how the extinction coefficients and the relative concentrations (ρ) were derived.
- 3) The authors conclude that all NEMO variants are highly photostable, but they only measured the photostability of one variant. As can be seen from the authors' data, photostability is highly dependent on the actual protein (mNG vs NEMOm), and such generalizations should therefore be omitted. The authors should rephrase this statement or provide photostability data for all NEMO sensors.
- 4) In Figure 2C experiment, the authors co-express the sensor with the relevant receptor. Did the authors confirm that receptor expression levels did not influence Ca²⁺ transients?
- 5) On line 47, the authors define DF/F₀ as the signal-to-background ratio (SBR). As noted by the authors, however, F₀ represents the sensor baseline, rather than true background fluorescence, and thus "signal-to-baseline ratio" would seem to be a more appropriate definition (e.g., Koldenkova, et al. BBA-Mol Cell Res, 2013). In fact, the latter definition is used throughout the rest of the manuscript. Please clarify this terminology.

Miscellaneous:

- Conflicting descriptions of the experiments performed in Figure 1B and C are given in the text (Line 78) and Figure legend (line 394). On line 78 the authors talk about using TG and EGTA and making use of the SOCE mechanism, but on line 394 they additionally mention ionomycin. Please clarify. If ionomycin was used it is unlikely that the SOCE mechanism is relevant to Ca²⁺ entry, and this should therefore be removed from the paragraph.
- Line 264-265: The authors first state that NEMOs has similar half decay time as GCaMP6s, but then immediately after state that the half decay time of NEMOs is slightly shorter. Perhaps mean to refer to two different NEMO variants here? Please clarify.
- Could the authors please comment on why they use a ratiometric readout for the photometry experiments, when their prior in vivo imaging was intensimetric?
- Please annotate Supplementary Table 2 to indicate which variants correspond to the final biosensors
- Supplementary Figure 5E: No errors and statistics are given even though the main text describes that the changes are not significant (line 269). Please add the errors and statistics to the figure.
- The authors often mention that N = 3, 14 cells per repeat. What is a repeat? Is it a separate preparation or a different FOV? Please clarify in the Material and Methods.
- The authors state in the Reporting Summary that no custom made code was used, but the Material and Methods section mentions MatLab code in multiple instances. This should be specified and the code made available if possible.

Author Rebuttal to Initial comments

Response to Reviewers' comments

Please note that changes made throughout the main text were shown in red.

Reviewer #1:

In this manuscript, Wang et.al., developed a series of mNeonGreen-based Ca²⁺ indicators having increased dynamic range with several accessory functions including high brightness and fast kinetics. Properties of NEMO are validated by well-designed in-vitro and in-cell analysis, demonstrating the high detectability of Ca²⁺ dynamics with improved SBR than those of existing GECIs. The authors also performed in-vivo analysis including 2-P microscopy and fiber photometry, clearly showing the potential utility of NEMO in living animals. Overall the manuscript is well-organized to emphasize the advantages of NEMO by focusing on the expanded dynamic range. However, the manuscript still has room for improvement in some data presentation, in which the lack of analysis details made reviewers hard to validate the potentially excellent work, and part of the logic is not simple and logical to draw a clear conclusion as pointed below.

We thank the reviewer for stating that our work is “*well-designed*”, and that our manuscript was “*well-organized*”.

Comments:

- 1) *Authors demonstrated the advantages of NEMO by primarily focusing on their increased DR which allowed the detection of Ca²⁺ dynamics with a larger signal change than existing indicators. In principle, the merits of expanded dynamic range should include a resolvability of similarly large signals which can be miss-evaluated to have the identical magnitude by indicators with a small dynamic range. Can authors strengthen this point?*

Response: We thank the reviewer for this excellent suggestion. We performed more *in cellulo* testing of NEMO sensors, added one new supplementary figure (Fig. S5) and the following description in the main text (Fig. R1). “We next examined whether the larger dynamic range of NEMO sensors could enable more sensitive detection of Ca²⁺ signals, which would otherwise appear to have similar magnitude of response due to the smaller dynamic range using existing GECIs. We compared the performance of NEMO sensors with GECIs bearing comparable Ca²⁺-binding affinities in response to small or large Ca²⁺ signals. We first tested the idea by resorting to an optogenetic tool, Opto-CRAC, which enables the stepwise induction of increased Ca²⁺ influx due to graded activation of endogenous ORAI Ca²⁺ channels by varying the photo-activation duration¹⁵⁻¹⁷. Opto-CRAC was subsequently co-expressed in HeLa cells along with NEMOm or GCaMP6m, respectively, for side-by-side comparison of photo-induced Ca²⁺ signals. Compared to GCaMP6m that poorly reported light-induced Ca²⁺ influx in small magnitudes, the NEMOm signals were significantly larger, with the amplitudes of NEMOm response showing stepwise increase in response to prolonged photo-stimulation from 100 ms, 300 ms to 1000 ms (**Fig. S5A**, and **Supplementary Video 3**). Second, we compared the SOCE responses of HEK293 cells when exposed to increasing extracellular Ca²⁺ concentrations. NEMOm or NEMOs could discriminate more external Ca²⁺ gradients than GCaMP6m or NCaMP7 did (**Fig. S5B-C**). Moreover, the signal-to-noise ratios (SNRs) of NEMOm and NEMOs were significantly higher than their corresponding counterparts, which may also contribute to their better resolvability over the amplitudes of Ca²⁺ signals (**Fig. S5D-F**). “.

Figure R1 (Fig. S5). NEMO performance in resolving Ca^{2+} signals that span a wide range of magnitude. (A) Dim blue light stimulation induced GECI responses in HeLa cells co-expressing Opto-CRAC and GCaMP6m or NEMOm. Left, typical traces; right, statistics. (B-C) SOCE indicated by GECIs in HEK cells bathed in stepwise external Ca^{2+} . (B) GCaMP6m vs NEMOm; (C) NCaMP7 vs NEMOs. Left, typical traces; right, statistics. (D-F) Statistics of panels A-C showing SNR of GECIs. $n = 3$ independent biological repeats.

2) When compared to parental NCaMP7, *in vitro* characterization (Table S4) showed that the increased dynamic range of NEMO was achieved by the decrease of min-brightness, not by an increase of max-brightness. Generally, low brightness causes the noisy baseline intensity that would spoil the signal-to-noise-ratio (SNR). Thus, a comparative analysis of SNR is essential to convince the utility of NEMO series in *in-cell* and *in-vivo* demonstrations. If NEMO shows better SNR than NCaMP7, it could be a strong point of this work to be emphasized.

Response: We thank the reviewer for this thoughtful suggestion. Our *in vivo* results showed that, even with GCaMP recording setup (excitation at 920 nm) that is suboptimal for NEMOs (most ideal excitation at 980 nm), the NEMOs' SNR was still significantly larger than GCaMP6s. This result indicates that NEMOs has a better *in vivo* SNR than NCaMP7, as it was shown to have an *in vivo* SNR similar to GCaMP6s (Fig. 6 in Subach et al, Int. J. Mol. Sci. 2020). In our original Fig. S7G, variance instead of Standard Deviation (SD) were mistakenly used to calculate the "SNR". Now we have updated it with correctly calculated SNR (Fig. S7G).

We also compared the performance of NEMOs and NCaMP7 in cell lines (Fig. R1C) and cultured neurons (Fig. R2A). The results showed that the sensitive version of our sensors, NEMOs, showed better SNR than NCaMP7 under most tested conditions. We added this data into new Supplementary Figures 5 and 7. Description of these results were incorporated into the main texts: In cultured neurons - "..... often with their SNR significantly larger than their counterparts"; *in vivo* - ".....appreciably better SNR (**Supplementary Fig. 7G**) and Probably due to its weak fluorescence, NEMOf signal obtained with GCaMP set up showed similar SNR to GCaMP6f (**Supplementary Fig. 7G**). Since the basal NEMOf fluorescence approximately doubled by switching from GCaMP excitation (920 nm) to a NEMO setup (980 nm) (**Supplementary Fig. 8C**), we thus collected NEMOf signals with 980 nm light. The results showed that NEMOf under optimized illumination retained its large SBR (**Supplementary Fig. 8D**, left panel), and the SNR of NEMOf was significantly better than that of GCaMP6f or NEMOf with 920nm excitation (**Supplementary Fig. 8D**, right panel). Since GCaMP6s and NCaMP7 were reported to have similar *in vivo* SNR⁷, it is likely that optimally-excited NEMOs (i.e., at 980 nm) may exhibit a better SNR than NCaMP7".

Fig. R2. The performance of NEMO sensors as indicated by the signal-to-noise ratios (SNRs) in neurons. A) Cultured neurons. B) *In vivo* (neurons in the mouse visual cortex).

We agree with the reviewer that lower basal brightness may compromise signal-to-noise-ratio (SNR). Our finding that NEMOs had generally larger SNRs than NCaMP7 is a surprise to us. We thus checked the origin of noises. In theory, the SNR can then be calculated by the following equation (<https://scientificimaging.com/knowledge-base/signal-and-noise-quantitative-explanation/>)

$$\text{SNR} = \frac{S}{\sqrt{S+N^2}} \quad (1)$$

Where S is the signal (in electrons) that contains photon shot noise; N is the system-dependent noise (in electrons).

Thus it is apparent that, when the brightness of the signal (S) is dim, the system dependent noise (N) would be a major contributor to the total noise levels, and greatly compromise the SNR. NEMOf is an example of sensors with dim basal fluorescence, about 1/4 of GCaMP6m and 1/8 of NEMOs in HEK293 cells (Fig. 1D). When tested *in vivo*, the basal NEMOm fluorescence doubled by switching from GCaMP excitation (920 nm) to a NEMO setup (980 nm) (Fig. R2B). We thus collected NEMOf signals with 980 nm light, the results showed that NEMOf under optimized illumination showed improved SNR, significantly larger than that of GCaMP6f or NEMOf when excited at 920 nm (Fig. S8D). These results agree with the general impression that lower brightness of a signal would follow by bad SNR. We have updated this in Fig. S8D, right panel.

However, under the condition that S is much larger than N and N could thus be neglected in the denominator of the equation, then Equation (1) could be simplified as:

$$\text{SNR} = \sqrt{S} \quad (2).$$

In this case, the SNR is mostly dependent on the signal, or shot noise. The larger the signal is, the bigger the SNR. We speculate that this is the case for NEMOs vs NCaMP7, as both of them have bright basal fluorescence (8- or 14-fold higher than that of NEMOf, respectively). Since NEMOs response was much larger that of NCaMP7, it would have a larger SNR (as seen in Fig. R1 F, Fig. S5F).

3) Authors could do a more straightforward presentation for NEMO's properties. It is notoriously confusing to round about the data between "in cell" and "in vitro" analysis for many variants. Concerning this issue, increased max-brightness of mNG-based indicator than GCaMP would be caused by the complete shifting of chromophore state to the bright anionic form. But this is not a unique feature of NEMO since NCaMP7 and related mNG-GECO already possessed this feature. Consequently, the uniqueness of NEMO might become the reduced brightness at low Ca²⁺ condition (and faster dissociation kinetics). To help readers' understanding of this point, data measured by the brightness would be needed especially for Fig1B and 1E in addition to F/F₀. Fig1B and 1D are not convincing in the

absence of the correction of the expression level of given indicators among cells and constructs. I also wonder why NEMO shows larger DR in cells than in vitro?

Response: We have followed the reviewer's excellent suggestion to better present our data, and also added the following sentences in the main text to better explain why NEMO sensors show larger dynamic ranges: "Consistent with their lower Ca^{2+} -binding affinities, the normalized basal brightness of all NEMO sensors were much lower than that of NCaMP7, and those of NEMOc or NEMOf were only about 0.25-0.5 of GCaMP6m. This finding indicates that the lower basal fluorescence of NEMO variants might contribute to the observed large dynamic range of NEMO indicators, especially for NEMOc or NEMOf. However, even though the dynamic ranges of NEMOm, NEMOs and NEMOb are over 5-fold higher than that of GCaMP6m, their basal fluorescence is either similar to or brighter than that of GCaMP6m (**Fig. 1D, 1F**). Hence, the high dynamic range for these three indicators may also be due to their maximal brightness being brighter than that of GCaMP6m."

We totally agree with the reviewer that a careful examination of brightness of GECIs is essential. And we indeed did correct the artifacts caused by variations in the expression levels of given indicators with "a P2A-based bicistronic vector to drive the co-expression of mKate (as an expression marker) and GECIs at a near 1:1 ratio" (Fig. 1F). By dividing the fluorescence of GECI (F_{GECI}) with that of mKate (F_{mKate}), the resulting ratio ($R = F_{\text{GECI}} / F_{\text{mKate}}$) would indicate the relative brightness of GECIs. To obtain the original Fig. 1F, we further normalized the ratios of GECIs against that of GCaMP6m to achieve the relative basal brightness of GECIs to that of GCaMP6m. To make the presentation more straightforward, we updated the Fig. 1F to just show normalized GECI ratios, and revised the related figure legend "F) Basal brightness of NEMO, NCaMP7 or GCaMP sensors. To achieve better estimation of the basal fluorescence of GECIs (F_{GECI}), F_{GECI} of cells expressing mKate-P2A-GECI constructs were normalized against the fluorescence of mKate, an expression marker (F_{mKate}). And the resulting ratio ($F_{\text{GECI}} / F_{\text{mKate}}$) was used to indicate the basal brightness of different GECIs."

Our purpose for Fig. 1B was to show the protocol of the screening assay for GECIs with larger dynamic ranges. To better illustrate the protocol and also to avoid confusion over the maximal fluorescence of GECIs, we updated a corrected trace for NCaMP7 (Fig. R3A). For NEMO data points shown in Fig. 1D and Fig. 1G, the values were indeed corrected for their expression levels, and therefore, their basal fluorescence truly reflects the brightness relative to GCaMP6m. We added the following description into the related figure legend: "To more accurately compare the performance of NEMO sensors with existing GECIs, the resting fluorescence of NEMO indicators in D) and G) were scaled with a factor F (calculated as $F = (\text{basal GECI ratio}) / (\text{rest GCaMP6m ratio})$)."

We also examined the dynamic ranges of these GECIs in the context of mKate-P2A-GECIs constructs (Fig. R3B-3D). The results revealed that, even though NEMO sensors still showed much better

performance than GCaMP6m or NCaMP7, their dynamic ranges were more or less compromised. This may be caused by the residual P2A peptide left on the N-terminus of GECIs after the cleavage of mKate-P2A-GECI at the P2A sites. We thus did not include this piece of data in the manuscript.

We do not know the exact reason as to “why NEMO shows larger DR in cells than *in vitro*”. We suspect that ion compositions or other factors that differ between these two experimental systems might contribute to this, which warrant further investigation in follow-on studies.

Fig. R3. The dynamic ranges of NEMO sensors with their expression levels corrected. A) Screening protocol shown by a typical NCaMP7 responses. Only baseline and SOCE responses were shown, and recordings of store depletion were not shown. B-D) NEMO performance with expression levels corrected. Prior to recordings, cells were bathed in store-emptying solution for 10min. To achieve more accurate estimation of the basal fluorescence of GECIs (F_{GECI}). F_{GECI} of cells expressing mKate-P2A-GECI constructs were normalized against that of an expression marker mKate (F_{mKate}). And the resulting ratio (F_{GECI} / F_{mKate}) was used to indicate the basal brightness of different GECIs. Prior to recordings, cells were bathed in 1 μ M TG for 10 min to fully deplete ER Ca^{2+} store and induce SOCE. Typical traces B) shown with fluorescence ratio; C) Shown by relative changes in ratio R/R_{min} . D) Statistics showing maximal ratio in the presence of 100 mM Ca^{2+} .

4) For *in-vivo* analysis, NEMO would have increased detectability of Ca^{2+} change than GCaMP6, but it is not convincing in the absence of comparative analysis of SNR. Also, readers may wonder whether NEMO shows higher performance than NCaMP7 and related mNG-GECO in detecting neuronal activities with expecting increased SBR, SNR and kinetics (and K_d). These could be tested at least for cultured neurons and hopefully for brain slices. Additionally, the optical properties of NEMO protein at two-photon excitation are informative.

Response: We followed the reviewer's advice, carried out more analysis on our existing results, and also performed more recordings in cell lines, cultured neurons and *in vivo*. The results showed that NEMOs or NEMOf showed increased SBR and SNR than their counterparts in cell lines and cultured neurons under most tested conditions (Fig R1 and Fig. R2A). Our *in vivo* results together with reports from literatures support the notion that NEMOs generally perform better than NCaMP7, with increased SBR or SNR and rapid kinetics (Fig. R2). The corresponding new figures were added into supplementary figures (Fig. S5C, S5F, S7A). Please also refer to our responses to comments 2 above for more details.

We have performed a series of new rigorous experiments to make comparisons among tested GECIs side-by-side (SNRs, SBRs).

- We compared SOCE responses of HEK293 cells exposed to increasing extracellular Ca^{2+} concentrations. Again, NEMOm or NEMOs showed significantly higher SNRs than GCaMP6m or NCaMP7 (Fig. S5E-F).
- We further recorded GECI responses in cultured neurons and compared the SNRs of GECIs. The results showed that NEMOs showed significantly higher SNRs than NCaMP7 and GCaMP6s under most tested conditions. Also, NEMOf showed higher SNR than GCaMP6f when the stimulus frequency was higher than 20 Hz.
- We also performed more *in vivo* recording with NEMOf (Fig. S8D) with a NEMO setup (980nm excitation), and NEMOf showed significantly better SNR than GCaMP6f or NEMOf with 920nm excitation (Fig. S8D, right panel). We speculate that NEMOs with 980 nm excitation may show further improved SNR. Because the reported *in vivo* SNRs of NCaMP7 and mNG-GECO are comparable to that of GCaMP6s, we reasoned that NEMOs sensor may also show better SNRs than these two indicators.
- We followed the reviewer's advice, and carried out *in vitro* two-photon characterization of NEMOc, checked the dynamic ranges of NEMOs under two-photon excitation in HEK293 cells (Fig. R4). The new data has been included in the new Fig. S3, along with the following description in the main text: "We also examined the *in vitro* normalized two-photon action cross-sections of NEMOc and *in cellulo* dynamic range of NEMOs under two-photon excitation when expressed in HEK293 cells. The properties and performance of NEMO sensors relative to GCaMP6m or NCaMP7 remained

largely similar to those observed with one-photon excitation (**Supplementary Fig. 3A–C**.)”

- We attempted for additional recordings on brain slices. However, the dilution effects of GECI after establishment of whole cell configuration was quite severe in our hands with our setup. The resulting basal fluorescence of GECIs was very dim. Even the kinetics and dynamic range of control GECI, GCaMP6f, did not quite match the published values. We therefore did not pursue this further. Regardless, we have successfully “*tested at least for cultured neurons*” as suggested by R1.

Fig. R4. (Fig S3). Two-photon spectral properties and performance of NCaMP7, GCaMP6m and NEMO sensors. A) Typical traces of normalized two-photon action cross-sections at pH 7.2. B) Dynamic ranges of the indicated GECIs when expressed in HEK293 cells. C) Statistics of basal fluorescence shown in panel B). n=3 independent biological repeats.

5) Application of NEMO in Fig2E is potentially excellent, but it is hardly evaluable in the absence of experimental details. From the viewpoint of K_d (528nM, *in vitro*), I also wonder about the reliability of the maximum or minimum concentration of Ca^{2+} in a cell. Provide detailed analysis schemes and discussion including in-cell calibration of Ca^{2+} affinity to ensure readers' understanding how iPEAQ using NEMO reliably determines Ca^{2+} concentration for 4microM at the peak and below 100nM at resting.

Response: We thank the reviewer for agreeing that the application of iPEAQ method “*is potentially excellent*”. The iPEAQ method can cancel out the artifacts caused by varying expression levels of GECIs, allowing measurements of Ca^{2+} levels, or at least comparing the differences in Ca^{2+} levels between treatments.

We thank the reviewer for reminding us to consider the reliable working range of NEMOf. We agree that the K_d of NEMOf makes it not a reliable reporter for Ca^{2+} levels lower than 100 nM or higher than 2 μ M. Thus NEMOf is not best suited to illustrate the full advantage of iPEAQ. We therefore replaced NEMOf with the more Ca^{2+} sensitive NEMOs, and carried out *in vitro* calibration and in-cell measurements (Fig. R5). The new results were shown in Fig. S6G and Fig. 2E, which further attest to the advantage of the iPEAQ method.

Fig. R5. Photochromic measurements of Ca^{2+} signals. A) *In vitro* Ca^{2+} titration curves of photochromism contrast $((\Delta F / F_0)_{hv})$, left panel) or fluorescence of NEMOs (right panel). B) Intermittent photochromism-enabled absolute quantification (iPEAQ) of Ca^{2+} signals in HEK293 cells.

We did try in-cell calibration on both NEMOf and NEMOs. However, the maximal fluorescence of NEMOf or NEMOs was always lower than the corresponding peak response induced by CCh or ionomycin, making it unusable for calibration. We suspect that this may be artifacts caused by some leakage of NEMO sensors in permeabilized cells or some interference of calibration solutions on NEMO indicators. We therefore did not pursue this further.

We also added more description of the iPEAQ measurements: “.....allows measurements of Ca^{2+} levels using photochromism contrast independent on GECI concentrations and the intensity of excitation light. Indeed, we found that NEMOf and NEMOs exhibited photochromic characteristics. Brief 405 nm UV illumination superimposed on 488 nm light could increase NEMOf fluorescence in an inversely Ca^{2+} dependent manner, with the peak fluorescence named as F_0 . After switching off the UV light, NEMOf quickly relaxed back to its basal state, or termed as the minimal fluorescence (F_{end}) (**Supplementary Fig. 6F**). One such photochromic cycle would allow the calculation of photochromism contrast, defined as $((F_0 - F_{\text{end}}) / F_0)_{\text{hv}}$. We then quantified Ca^{2+} release induced by submaximal stimulation of muscarinic acetylcholinergic receptor with 10 μM CCh by taking the iPEAQ approach. Briefly, based on a Ca^{2+} -titration curve of photochromism contrast (**Supplementary Fig. 6G**, left), we converted the measured rest photochromism contrast of a NEMOs-expressing cell to basal Ca^{2+} concentration. With this calculated resting Ca^{2+} level and the Ca^{2+} -NEMOs fluorescence response curve (**Supplementary Fig. 6G**, right), the NEMOs fluorescence response of each cell could then be calculated as changes in the absolute Ca^{2+} concentrations (**Fig. 2E**).”

6) *Technical comments on Ca^{2+} titration: Are there any reasons to utilize HEPES? Experts in this field routinely utilize photostable MOPS rather than photo-unstable HEPS, since the latter is producible with H_2O_2 upon light illumination that can directly affect the property of indicators and indirectly affect the free Ca^{2+} concentration through the reduced buffering ability.*

Response: We thank the reviewer for pointing this issue out. The NEMO sensors were generated from NCaMP7, and NCaMP7 was characterized in HEPES buffer. We thus utilized HEPES buffer, so that we can directly compare the parameters of NEMO sensors with those of NCaMP7. The affinity of GCaMP6m measured in HEPES was 174 ± 6 nM (Table S3), which remained comparable to its reported values (167 ± 3 nM) obtained with MOPS (Chen et al, Nature, 2013), indicating that the potential issue caused by photo-unstable HEPES was rather minimal with our experimental setup.

7) *Please provide precise descriptions to avoid any ambiguities;*

(a) *L127: What is the clear answer for “Therefore, the larger dynamic range of NEMO indicators is not a mere reflection of lower basal fluorescence.” ? (see also comment (c))*

Response: We apologize for this ambiguity in our description. The larger dynamic range could result from one or the combinations of the following factors: a) Lowering of minimal fluorescence (All NEMO

sensors vs NCaMP7, NEMOf/c vs GCaMP6m); b) Brightening of the maximal fluorescence (NEMO sensors vs GCaMP6 indicators). We removed the sentence to avoid distraction.

(b) L141-: specify the reason for comparing the molecular brightness in the presence of Mg²⁺.

Response: We thank the reviewer for pointing this out. We have added the following sentence in the related table legend. “The effect of Mg²⁺ was tested because the free cytosolic Mg²⁺ concentration is in the range of mM and the presence of Mg²⁺ might affect the dynamic range of NCaMP7 (Table 1 in Subach et al, 2020), the parental template of NEMO sensors.”

(c) L147: “superior dynamic range of NEMO indicators can be explained by Ca²⁺-induced changes in both the proportion and the absolute brightness of the anionic fluorophores of NEMO variants.” This statement is true when compared to GCaMP6, but is false when compared to NCaMP7.

Response: We thank the reviewer for pointing this out, and have rephrased the sentence as follows: “...be largely ascribed to the Ca²⁺-dependent fold-of-increase in the molecular brightness of the anionic fluorophores of NEMO variants”.

(d) L155: specify the basis for 40x (FigS3 left) and 60x (FigS3 right) by referring appropriate Supplementary Figs, respectively.

Response: We followed the reviewer’s advice and added more description: “It endured nearly 40 times (1.518 mW) higher illumination than GCaMP6m (0.038 mW), and showed no apparent photobleaching. The stronger illumination (from 0.038 mW to 1.518 mW) could potentially enhance the basal fluorescence of NEMOm sensor by over 60-fold”.

(e) L158: specify the scenario requiring “strong illumination”.

Response: As shown in our *in vivo* measurements, NEMOf is a dim but fast sensor, suitable for imaging fast-acting neural activities. And brighter NEMOf fluorescence achieved by optimal illumination was

accompanied by better SNR (Fig. S8C and the right panel Fig. S8D). Similarly, brighter NEMOf fluorescence achieved by stronger illumination may also be helpful to obtain better SNR. Dim signal with better SNR is especially needed for the imaging of subcellular compartments, like dendrite spines. We thus added the following sentence in the main text: ", such as the *in vivo* monitoring of Ca²⁺ signals within subcellular compartments with dim NEMOf indicator"

(f) L207: "directly indicating basal Ca2+ level" is not logical.

Response: We agree with the reviewer and changed the phrase to "capable of indicating absolute Ca²⁺ level".

(g) L158: It is overstating that "collectively~ NEMO variants as highly photo-stable sensors", in the absence of comparative photobleaching analysis at saturating Ca2+ conditions.

Response: We agree with the reviewer and have toned down our conclusion by removing "highly".

(h) correct typos in SI;

-Data analysis section: the peak signal to ratio (SNR): "noise" is missing

-Legend for Supplemental Figure: Fig4. E) Left----Left.... "Right" is missing

Response: We apologize for these typos and have added the missing "noise" in the data analysis section, and changed to "Left, Middle, Right, statistics." in the figure legend for Fig. S6E.

Reviewer #2:

None

Reviewer #3:

Jia Li and colleagues report a new suite of GECIs called NEMO, or mNeonGreen-based calcium indicators, which are derived from NCaMP7 by modifying linker sequences, introducing calmodulin mutations, or including a nuclear export signal. All of these new NEMO sensors are shown to display extreme enhancements in dynamic range, both in vitro and in 293 cells, versus current sensors (e.g., GcaMP6, jGCaMP8), with several hundred-fold changes in fluorescence intensity between apo and Ca²⁺-bound states. The authors show that NEMO very robustly detects receptor-stimulated Ca²⁺ elevations, including GPCR-mediated oscillations. In cultured neurons, NEMO compares favorably with current GECIs in detecting single action potentials and dwarfs current best-in-class GECIs in response to increasing stimulation frequencies. Finally, the authors use NEMO to robustly visualize Ca²⁺ responses in the brains of awake mice via 2p imaging in the V1 visual cortex and fiber photometry in the corpus striatum. The author's work is thorough, with careful characterization experiments and benchmarking against relevant sensors that are currently in wide use. NEMO is undoubtedly an impressive new tool that will be of great interest and utility.

We thank the reviewer for considering our work as “*thorough*”, and for stating that our sensor set is “*undoubtedly an impressive new tool that will be of great interest and utility.*”

Specific comments:

- 1) *In Figures 4 and S5, the authors show two-photon imaging data acquired from acute brain slices and the brains of awake mice to claim improved performance of their NEMO sensors for in vivo imaging. The authors are strongly encouraged to add in vitro 2p characterization of their sensors, similar to 1p characterization in Figures 1 and S2 to enable more robust and absolute evaluation of 2p performance.*

Response: We thank the reviewer for this constructive suggestion. We carried out *in vitro* two-photon characterization of NEMOc, re-characterized the dynamic range of NEMOs under two-photon excitation in HEK293 cells (Fig. R4), and added the new data in Fig. S3. Please refer to our responses to the Comment 4 of R1 above for more details.

- 2) *Lines 135-149: While the overall conclusions drawn by the authors seem consistent, it is hard to follow the experiments performed to differentiate the proportion vs absolute brightness. Neither this paragraph, nor Supplementary Figure 2, Supplementary Table 4 or the Material and Methods describe the experiments and data analysis in sufficient detail for the reader to follow. Please include a more precise description of how the extinction coefficients and the relative concentrations (ρ) were derived.*

Response: We apologize for not describing our results in a more straightforward way. We have added labels on Fig. S2D to show key parameters needed for the quantification, and a paragraph in the method section to describe a stepwise protocols used to derive the extinction coefficients and the relative concentrations (ρ): “Briefly, the corresponding absorbance (OD) in protonated (OD_N , peak absorption at ~ 400 nm) and deprotonated (OD_A , peak absorption at ~ 500 nm) states were then obtained from the absorption spectrum curves. Next, we performed linear regression on $OD_N - OD_A$ values at pH = 7, 7.2, 8, and 9 to obtained the slope S

$$S = \Delta OD_A / \Delta OD_N \quad (3)$$

The relationship between the concentration of a chromophore (n), its absorbance (OD) and extinction coefficients (ϵ) is:

$$n = OD / \epsilon \quad (4)$$

For alkaline-denatured ($_D$), protonated ($_N$) and deprotonated ($_A$) chromophores, the corresponding equation would be $n_D = OD_D / \epsilon_D$, $n_A = OD_A / \epsilon_A$ and $n_N = OD_N / \epsilon_N$. And green GECIs will be totally denatured at pH 12.5, thus $n = n_D$. Also the concentration of total chromophore (n) will not change during H^+ titration, hence

$$n_D = n_A + n_N \quad (5)$$

and

$$\Delta n_A = -\Delta n_N \quad (6)$$

Putting equation (4) or its variants into (5) and (6) would make:

$$OD_D / \epsilon_D = OD_A / \epsilon_A + OD_N / \epsilon_N \quad (7)$$

$$\Delta OD_A / \epsilon_A = -\Delta OD_N / \epsilon_N \quad (8)$$

ϵ_D of denatured GFP-like chromophores ($44,000 \text{ M}^{-1} \text{ cm}^{-1}$) was taken as ϵ_D of green GECIs at pH 12.5¹.

And equation (8) can be rearranged as: $\epsilon_A / \epsilon_N = -\Delta OD_A / \Delta OD_N$ (9)

Combining equation (3) and (9), we will obtain

$$\epsilon_A / \epsilon_N = -S \quad (10)$$

By solving equations (5) and (8), we can get

$$\epsilon_N = \frac{\frac{OD_A + OD_N}{-S}}{OD_D / \epsilon_D} \quad (11)$$

$$\varepsilon_A = \varepsilon_N * (-S) \quad (12)$$

And

$$\rho_A = \frac{n_A}{n_A+n_N} = \frac{\frac{OD_A}{\varepsilon_A}}{\frac{OD_A}{\varepsilon_A} + \frac{OD_N}{\varepsilon_N}} \quad (13)$$

$$\rho_N = \frac{n_N}{n_A+n_N} = \frac{\frac{OD_N}{\varepsilon_N}}{\frac{OD_A}{\varepsilon_A} + \frac{OD_N}{\varepsilon_N}} \quad (14)$$

- 3) *The authors conclude that all NEMO variants are highly photostable, but they only measured the photostability of one variant. As can be seen from the authors’ data, photostability is highly dependent on the actual protein (mNG vs NEMOm), and such generalizations should therefore be omitted. The authors should rephrase this statement or provide photostability data for all NEMO sensors.*

Response: We have followed the reviewer’s excellent advice to examine the photostability of all NEMO sensors. The results showed that all NEMO indicators turned out to be more photostable than GCaMP6m (Fig. R6). The new data were updated in revised Fig. S4A (bottom panels). We also toned down our conclusion by changing “highly photostable” to “photostable”.

Fig. R6. Statistics showing the relative reduction in fluorescence at the end of 200 sec illumination with dim (left) or strong (right) illumination.

- 4) *In Figure 2C experiment, the authors co-express the sensor with the relevant receptor. Did the authors confirm that receptor expression levels did not influence Ca2+ transients?*

Response: BmGr-9 was tagged with mScarlet, and the expression level of BmGr-9 was indicated by the mean intensity of mScarlet fluorescence in BmGr-9 expressing cells. NEMOs or GCaMP6s expressing cells showed similar levels of mScarlet fluorescence, indicating that the larger NEMOs response was not caused by the difference in BmGr-9 expression (Fig. R7).

Fig. R7. In HEK293 cells co-expressing NEMOs or GCaMP6s with BmGr-9-mScarlet, the expression levels of BmGr-9 were indicated by mean mScarlet fluorescence in individual cells.

5) *On line 47, the authors define DF/F_0 as the signal-to-background ratio (SBR). As noted by the authors, however, F_0 represents the sensor baseline, rather than true background fluorescence, and thus “signal-to-baseline ratio” would seem to be a more appropriate definition (e.g., Koldenkova, et al. BBA-Mol Cell Res, 2013). In fact, the latter definition is used throughout the rest of the manuscript. Please clarify this terminology.*

Response: We have corrected the full name of SBR as “signal-to-baseline ratio” as advised and clarified this terminology throughout the revised manuscript.

Miscellaneous:

- *Conflicting descriptions of the experiments performed in Figure 1B and C are given in the text (Line 78) and Figure legend (line 394). On line 78 the authors talk about using TG and EGTA and making use of the SOCE mechanism, but on line 394 they additionally mention ionomycin. Please clarify. If ionomycin was used it is unlikely that the SOCE mechanism is relevant to Ca^{2+} entry, and this should therefore be removed from the paragraph.*

Response: We thank the reviewer for noticing the discrepancy in our description and the usage of ionomycin. Ionomycin is an ionophore that could permeabilize membrane in a concentration dependent manner. At 10 μ M, ionomycin could permeabilize both ER and PM, causing nonspecific Ca^{2+} release and entry. Thus 10 μ M ionomycin is routinely used for the calibration of Ca^{2+} indicators. At 2.5 μ M, ionomycin is also known to deplete ER Ca^{2+} store to induce SOCE. We applied both 2.5 μ M ionomycin and 1 μ M TG to ensure that the ER Ca^{2+} store was fully depleted. We did use the SOCE mechanism to obtain maximal signals (F_{max}) of GECIs. We did not use 10 μ M ionomycin for this experiment because

cells permeabilized with 10 μ M ionomycin were not very healthy and tended to float away during solution changes.

We have added the description about ionomycin in the main text. “....., 2.5 μ M ionomycin (iono) and 1 μ M thapsigargin (TG) for 10 min. Ionomycin is an ionophore and TG is an inhibitor of the sarcoplasmic/endoplasmic reticulum Ca²⁺ ATPase, both of which serve as inducers of passive ER store depletion.”

- *Line 264-265: The authors first state that NEMOs has similar half decay time as GCaMP6s, but then immediately after state that the half decay time of NEMOs is slightly shorter. Perhaps mean to refer to two different NEMO variants here? Please clarify.*

Response: We thank the reviewer for pointing this out. The decay time of NEMOm (1.5 ± 0.2 s) was similar to that of GCaMP6s. We have corrected the sentence as follows: “Consistent with *in vitro* characterization, NEMOm was relatively slow, with the half decay time similar to that of GCaMP6s (1.36 ± 0.08 s)”

- *Could the authors please comment on why they use a ratiometric readout for the photometry experiments, when their prior in vivo imaging was intensimetric?*

Response: Ratiometric readout was recently developed for photometric measurements with GCaMP6 sensors (Martianova et al, 2019, JOVE). The dim GCaMP6 fluorescence excited at 410 nm remains largely unresponsive to changes in Ca²⁺ levels, making it a good indicator for the expression level of GCaMP6 and motion artifacts occurring in moving animals. We cited the reference and added the following sentence in the related method section: “GECI signal arising from 410 nm excitation was used as a Ca²⁺-independent reference to cancel out motion artifacts”.

- *Please annotate Supplementary Table 2 to indicate which variants correspond to the final biosensors.*

Response: We have highlighted variants corresponding to the final NEMO sensors in **yellow** in Table S2.

- *Supplementary Figure 5E: No errors and statistics are given even though the main text describes that the changes are not significant (line 269). Please add the errors and statistics to the figure.*

Response: We have replaced it with a correct one.

- *The authors often mention that $N = 3$, 14 cells per repeat. What is a repeat? Is it a separate preparation or a different FOV? Please clarify in the Material and Methods.*

Response: We have replaced all “ $N = 3$ ” with “ $N = 3$ independent biological replicates”. The cell numbers for each repeat were also indicated when appropriated.

- *The authors state in the Reporting Summary that no custom made code was used, but the Material and Methods section mentions MatLab code in multiple instances. This should be specified and the code made available if possible.*

Response: We mostly used customized Matlab scripts for very basic matrix operations, such as extracting data points from excel spread sheets that contain raw data, calculations, and commonly used statistical analysis. The related codes can be shared upon reasonable requests. We have added the following sentence in the related method section: “The Matlab codes are available upon request”.

Decision Letter, first revision:

Our ref: NMETH-A49501A

23rd Dec 2022

Dear Dr. Wang,

Thank you for submitting your revised manuscript "Engineering of NEMO as calcium indicators with ultra-high dynamics and sensitivity" (NMETH-A49501A). As I mentioned before, it has been seen by the original referees and their comments are below. The reviewers find that the paper has improved in revision, and therefore we'll be happy in principle to publish it in Nature Methods, pending minor revisions to satisfy the referees' final requests and to comply with our editorial and formatting guidelines.

Specifically, we ask that you explain the limitations of the dynamic range measurements in the manuscript.

We are now performing detailed checks on your paper and will send you a checklist detailing our editorial and formatting requirements in about 2-3 weeks. Please do not upload the final materials and make any revisions until you receive this additional information from us.

TRANSPARENT PEER REVIEW

ORCID

Sincerely,

Nina Vogt, PhD
Senior Editor
Nature Methods

Reviewer #1 (Remarks to the Author):

The majority of my concerns were appropriately addressed and corrected in the revised manuscript. I, however, express serious concerns about the author's response to my Q#3 related to the revised Fig1

and its associated texts.

A. Procedure of the data normalization.

Please check the validity of the data normalization. I found that the imaging condition for GCaMP was different from that for NEMO and NCaMP7, where the former would be excited with twice larger power due to the inhomogeneous spectrum of the light source (<https://www.optoscience.com/maker/excelitas/pdf/plugin-LDGI-XCite-RelativeOutput-120Q-vs-HBO100.pdf>). Under such conditions, I believe that a more careful normalization is essential to compare the min and max brightness of NEMOs and GCaMP (revised Fig. 1F and Fig. R3D). Similarly, "To more accurately compare the performance of NEMO sensors with existing GECIs, the resting fluorescence of NEMO indicators in D) and G) were scaled with a factor F (calculated as $F = (\text{basal GECI ratio}) / (\text{rest GCaMP6m ratio})$)." would not be accurate.

B. Data normalization by P2A system

Compared with the original Fig. 1B and C, Fig. R3 B and C are well organized to clearly show that an expanded DR of NEMO was achieved by lowered basal brightness while preserving max brightness of NCaMP7. However, the authors decided "not to include this piece of data in the manuscript" by considering that "even though NEMO sensors still showed much better performance than GCaMP6m or NCaMP7, their dynamic ranges were more or less compromised." I don't agree with this in the absence of a reasonable explanation for "compromised DR". Rather, authors should more carefully check the possibility that over-estimation of in-cell DR and under-estimation of in vitro DR as detailed in my next comment (C). Also, the authors' speculation that "This may be caused by the residual P2A peptide left on the N-terminus of GECIs after the cleavage of mKate-P2A-GECI at the P2A sites" is not convincing. If this is the case, "GECI>P2A>mKate constructs" in which residual P2A peptide emerges C-term of GECIs should be tested.

C. Inconsistency between in-cell and in-vitro DR

In general, in-cell DR has been attenuated due to the functional interference of GECI's CaM and M13 by endogenous CaM and its binding targets (PMID: 16720273). Since many experts in this field no doubt wonder why NEMO shows larger DR in-cell than in vitro, a reasonable explanation would be essential. Although I do not deny the possible involvement of a different ion composition, a more logical discussion should be included in the main text by citing past papers reporting larger DR of GECIs in-cell than in vitro (if available).

More importantly, inappropriate data normalization and the lack of reasonable explanation about exceptionally large in-cell DR forced me to be suspicious about the validity of the presented values of DR. As an expanded DR of NEMO is the main achievement of the manuscript, please clarify the following points for the possible involvement of measurement and/or calculation artifacts.

(C1) Over-estimation of in-cell DR

Biased cell sampling: It has been known that the magnitude of Ca²⁺ responses can be affected by the dosage of Ca²⁺ reporter (PMID: 14872098). Please deny the possibility of biased cell sampling with too small loading of GECIs especially for NEMOc and NEMOm. In this context, notice that normalization of GECI's intensity by reference intensity of mKate is useless.

Calculation artifacts: Exceptionally large DR (in-cell) values could be obtained if too small denominator was used. Please carefully check the appropriateness of your image processing by considering over-subtraction of background intensity, undesired effects of the cutoff function commonly pre-installed to sCMOS sensors, and other possible calculation artifacts associated with uniquely introduced

procedures such as "To avoid saturation of the camera, after recording the rest fluorescence (F0) with regular exposure time (approximately 500 ms), time-series for variants with high dynamic range were recorded using one-tenth to one-fifth the exposure time. Afterwards, the fluorescence response curves of each cell were scaled up according to the corresponding F0."

If raw image data is available, I am willing to help check these issues.

(C2)Under-estimation of in vitro DR

Please check your experimental conditions for in vitro measurement, if in-cell DR is found to be free from any artifacts. The in vitro performance of NEMO could be attenuated if a detector's DR and linearity were not appropriately tuned. For a PMT detector with a small DR, a signal saturation at the high-intensity range would occur to spoil the DR of NEMO. To avoid this, authors might utilize a high-dynamic range mode of "Flexstation 3, plate reader" allowing an exceptionally wide range of intensity measurement from 10^1 to 10^6 counts. Generally, such a wide-range detection is realized by a dynamic combination of three different PMT sensitivities (high-middle-low) for which their linearity is not always guaranteed.

(C3) No artifacts

When authors find DR values are free from any artifacts, a logical and reasonable discussion explaining a large mismatch btw in-cell DR and in vitro DR is needed.

In summary,

- (1) I have to say that the inconsistency of DR is unacceptable because the expanded DR is the key property of the newly generated NEMO series. (General expectation; in vitro DR>in-cellDR)
- (2) Data normalization should be appropriately performed for Fig1C, 1D, 1F, and 1G.
- (3) For Fig1E or 1C, I believe that it is highly informative to show associated intensity data (at least in the Supplementary section) since it allows readers' at-a-glance understanding of why the DR of NEMOs is expanded.

Further corrections;

- DR is generally defined by $\Delta F/F_{min} = (F_{max} - F_{min})/F_{min}$, rather than F_{max}/F_{min} (line 77 and others).
- Mixed usage of "basal" and "resting" such as " $F = (\text{basal GECI ratio})/(\text{rest GCaMP6m ratio})$ at L494" is confusing. Specify their differences if these were not identical ones.
- Be consistent with the distinction between "R0 and Rmin" or "F0 and Fmin". F0 on the y-axis in Fig1D would be " F_{min} (F at zero-Ca²⁺)".
- Nuclear Export Signal is common rather than "exclusion" in Fig 1A.
- 405nm is not UV but violet light.
- If available, cite appropriate references for plasmodesmata-associated Ca²⁺ signals to allow readers can judge whether your findings are consistent with previous reports. If this were a pretty new finding, add a reasonable explanation for this finding.
- Make boxes and error bars to be clearer in Fig1F.
- L33: Make sure the validity and consistency of DR values (>200-fold, L33 vs >100-fold, L108).
- L63: Update ref10 to PMID: 32571014

Reviewer #3 (Remarks to the Author):

The authors should be applauded for their thoroughness. All reviewers' concerns have been

appropriately addressed. Regarding the in vitro 2P characterization added to Figure S3, it would be nice if the authors could also provide the molecular two-photon brightness values for the examined sensors.

Author Rebuttal, second revision:

Reviewer #1:

The majority of my concerns were appropriately addressed and corrected in the revised manuscript. I, however, express serious concerns about the author's response to my Q#3 related to the revised Fig1 and its associated texts.

Response: We thank the reviewer #1 for giving us insightful and constructive suggestions. These comments help us to better estimate the dynamic ranges (DRs) of NEMO sensors.

A. Procedure of the data normalization.

Please check the validity of the data normalization. I found that the imaging condition for GCaMP was different from that for NEMO and NCaMP7, where the former would be excited with twice larger power due to the inhomogeneous spectrum of the light source (<https://www.optoscience.com/maker/excelitas/pdf/plugin-LDGI-XCite-RelativeOutput-120Qvs-HBO100.pdf>). Under such conditions, I believe that a more careful normalization is essential to compare the min and max brightness of NEMOs and GCaMP (revisedFig.1F and FigR3D). Similarly, "To more accurately compare the performance of NEMO sensors with existing GECIs, the resting fluorescence of NEMO indicators in D) and G) were scaled with a factor F (calculated as $F = (\text{basal GECI ratio}) / (\text{rest GCaMP6m ratio})$)." would not be accurate.

Response: We agree with the reviewer that it is difficult to equalize the imaging conditions between GCaMP6m and NEMO sensors, and that our scaling procedure would cause confusion.

With regard to the updated Fig. 1F, we now show raw readings of GECIs obtained with the same imaging setups: either with a GFP filter set (GFP fluorescence) or with a YFP filter cube (YFP fluorescence). To eliminate artifacts caused by different expression levels of GECIs, a P2A system was used and mKate fluorescence (F_{mKate}) was used as an indicator of expression levels. The basal fluorescence (F_{GECI}) was thus indicated as fluorescent ratios

($F_{\text{GECI}} / F_{\text{mKate}}$) (Fig. R1A).

For the updated Fig. 1D and 1G, we now showed un-normalized raw readings of basal fluorescence. We included herein a replot of the original Fig. 1G using raw basal readings (Fig. R1B). To obtain better estimation of in-cell DR, we carried out new measurements using a YFP-filter cube to collect fluorescence of all GECIs (Fig. 1G). To more accurately obtain minimal (F_{min}) and maximal fluorescence (F_{max}), we first defined the linear range and generated a calibration curve (exposure time vs fluorescence) of our camera (Fig. R1C), then obtained raw F_{max} with shorter exposure time (1/4X or 1/2X) to avoid camera saturation, recorded raw F_{min} with a longer exposure time (10X) to minimize artifacts from “cut off” function of the camera. We finally calculated F_{min} , F_{max} and the corresponding DR values using the calibration curve of the camera (left panel in Fig. R1C, Fig. R1D-R1F).

Both replotted or newly obtained data showed similar trend to those previously shown.

Figure R1. Basal fluorescence (F_0) and dynamic range (DR) of GECIs. A) Basal brightness of NEMO, NCaMP7 or GCaMP sensors viewed with YFP (top) or GFP (bottom) filters. To achieve better estimation of the basal fluorescence of GECIs (F_{GECI}), F_{GECI} of cells expressing mKate-P2A-GECI constructs were normalized against the fluorescence of mKate, an expression marker (F_{mKate}). B) Replots of the original Fig. 1G, F_0 -dynamic range of individual cells expressing NEMO variants or GCaMP6m. Raw F_0 readings were used. C) Exposure time vs NEMO fluorescence curve obtained with

the sCMOS camera used in this study. Left panel, individual traces; right panel, traces normalized against readings obtained with 1000 ms exposure (mean \pm SEM). D) Typical traces used to calculate dynamic ranges of the indicated GECIs using store-emptied HEK293 cells. E) A new set of F_0 -dynamic range of individual cells. Raw F_{\max} was obtained with shorter exposure time (1/4X or 1/2X) to avoid camera saturation, raw F_{\min} was recorded with a longer exposure time (10X) to minimize artifacts from “cut off” function of the camera. F_{\min} , F_{\max} and the corresponding DR values were then calculated using calibration curve in C). F) Statistics.

B. Data normalization by P2A system

Compared with the original Fig1B and C, FigR3 B and C are well organized to clearly show that an expanded DR of NEMO was achieved by lowered basal brightness while preserving max brightness of NCAMP7. However, the authors decided “not to include this piece of data in the manuscript” by considering that “even though NEMO sensors still showed much better performance than GCaMP6m or NCaMP7, their dynamic ranges were more or less compromised.” I don’t agree with this in the absence of a reasonable explanation for “compromised DR”. Rather, authors should more carefully check the possibility that overestimation of in-cell DR and under-estimation of in vitro DR as detailed in my next comment (C). Also, the authors’ speculation that “This may be caused by the residual P2A peptide left on the N-terminus of GECIs after the cleavage of mKate-P2A-GECI at the P2A sites” is not convincing. If this is the case, “GECI>P2A>mKate constructs” in which residual P2A peptide emerges C-term of GECIs should be tested.

Response: We agree with the reviewer that our previous speculation regarding the possible effects of residual P2A peptides on DRs of GECIs lacks experimental evidence. As detailed below, our P2A system did contain small but significant “contamination” signal from unmaturing mKate, resulting in less accurate DR values. Thus DR values obtained with the P2A system are prone to have more calculation artifacts.

For our P2A system, a red fluorescence protein mKate was used as a reference for protein expression. We did not use BFP or CFP, as their excitation lights would interfere with the fluorescence of the photochromic NEMO sensors (Fig. S6E). It is known that during maturation, red fluorescence protein would yield green fluorescence (Fig. R2A, around 5%). This small

contamination became significant when measuring those small minimal fluorescence values (Fig. R2B&C), reducing the measured DRs of GECIs. Since the maturation rate of mKate varies from cell to cell, calibration and subtraction of this “contamination” is flawed by nature. Over-subtraction would increase the apparent DR, while sub-subtraction would decrease the apparent DR, resulting in “calculation artifacts”.

Indeed, that was what we observed (Fig R2B&C). The DR values obtained with the P2A system were generally smaller, likely caused by incomplete correction of contamination from un-matured mKate (Fig. R2C). Also, a portion of DRs obtained with the P2A system were much larger, especially for those of NEMOc (Fig. R2C), probably due to over-subtraction of “contamination” from un-matured mKate. We hope that the reviewer would now agree with us for not including DR data obtained with the P2A system.

Figure R2. Minimal fluorescence (F_{min})- DR plots of GECIs. The performance of cells expressing GECIs (GECI only), or mKate-P2AGECIs (GECI in the P2A system) was compared. A) Full scale; B) Smaller scale showing the performance of GCaMP6m, NEMOm and

C. Inconsistency between in-cell and in-vitro DR

In general, in-cell DR has been attenuated due to the functional interference of GECI's CaM and M13 by endogenous CaM and its binding targets (PMID: 16720273). Since many experts in this field no doubt wonder why NEMO shows larger DR in-cell than in vitro, a reasonable explanation would be essential. Although I do not deny the possible involvement of a different ion composition, a more logical discussion should be included in the main text by citing past papers reporting larger DR of GECIs in-cell than in vitro (if available).

More importantly, inappropriate data normalization and the lack of reasonable explanation about exceptionally large in-cell DR forced me to be suspicious about the validity of the presented values of DR. As an expanded DR of NEMO is the main achievement of the manuscript, please clarify the following points for the possible involvement of measurement and/or calculation artifacts.

Response: As listed below, we carefully carried out additional better-controlled experiments, re-analyzed existing data, and added discussions, to address the reviewer's concern.

(C1) Over-estimation of in-cell DR

Biased cell sampling: It has been known that the magnitude of Ca²⁺ responses can be affected by the dosage of Ca²⁺ reporter (PMID: 14872098). Please deny the possibility of biased cell sampling with too small loading of GECIs especially for NEMOc and NEMOm. In this context, notice that normalization of GECI's intensity by reference intensity of mKate is useless.

Response: As shown in Fig. R1B, the updated Fig 1G or Fig R1E, the DR values of NEMOm are relatively stable over a wide range of basal fluorescence intensities. Here we replotted the raw NEMOm signals (Fig. R3A). As can be seen with linear regression, higher expression of the GECI (higher basal fluorescence) did not seem to significantly reduce DR of NEMOm (The slope is not significantly non-zero).

With regard to NEMOc, it is difficult to address dosage effect as higher basal reading (larger than 230 a.f.u.) may lead to its saturation at high Ca²⁺ levels. From the background-subtracted raw traces of NEMOc (Fig R3B and R3C), it is clear that, with similar basal readings to that of GCaMP, the maximal NEMOc signal got "cut off" on the top due to saturation of the camera.

Based on the fact that NEMOm showed little dosage effect, we would speculate NEMOc would also have minimal dosage effect.

Figure R3. The dosage effects of NEMO sensors. A) Plots of rest fluorescence (F_0)-DR. B-C) Responses of selected cells with comparable NEMO or GCaMP6m fluorescence.

Calculation artifacts: Exceptionally large DR(in-cell) values could be obtained if too small denominator was used. Please carefully check the appropriateness of your image processing by considering over-subtraction of background intensity, undesired effects of the cutoff function commonly pre-installed to sCMOS sensors, and other possible calculation artifacts associated with uniquely introduced procedures such as "To avoid saturation of the camera, after recording the rest fluorescence (F_0) with regular exposure time (approximately 500 ms), time-series for variants with high dynamic range were recorded using one-tenth to one-fifth the exposure time. Afterwards, the fluorescence response curves of each cell were scaled up according to the corresponding F_0 ."

If raw image data is available, I am willing to help check these issues.

Response: With our imaging setup (bin=2), the offset (or cutoff) of our camera is 400, close to the averaged background reading that was around 425 for NEMOc/NEMOm, and 485 for GCaMP6m (Fig. R4A). Thus even though small in numbers, there might be over-subtraction of background intensity, leading to some overestimation of DR values.

Originally, to measure DR that is large (200~400 folds), we had to keep minimal fluorescence (F_{\min}) values low (For NEMOc, mean F_{\min} =100 afu, Fig. R1B, left panel), as we started to see camera saturation in NEMOc expressing cells with F_{\min} >170 afu (Fig. R3B). For cells with high expression of NEMOc, if we got rid of these saturated cells, then only cells with low DR values left, resulting in a "trend" of decreasing DR with brighter NEMOc basal

fluorescence (Fig. 4B). And as also stated by the reviewer, such small F_{\min} values are prone to calculation artifacts. Thus it is really difficult to eliminate calculation artifacts in measuring in-cell DRs that are over 200 fold.

We thus carried out new in-cell DR measurements, and minimized calculation artifacts by obtaining more reliable readings of F_{\min} (Fig. R1 E&F). (Please refer to our responses to question 1 for details). The results showed that the DR of GCaMP6m is 15.3 ± 3.6 . The DR of NEMOc (377.8 ± 10.9) is more than 23-folds higher, those of NEMOm/f (203.1 ± 4.6 , 213.8 ± 5.1 , respectively) are more than 12.3 folds higher, and that of NEMOs (85.1 ± 1.7) is more than 4.5 folds higher. Therefore, even though the newly-measured DRs of all GECIs were slightly smaller, the fold-of-increase in DR values relative to GCaMP6m were still similar to those previously obtained ones.

We also replotted data from original Fig. 1B, showing DRs of cells with comparable F_{\min} (Fig. R4B). There is no doubt that NEMOc- or NEMOm-expressing cells still showed much larger DR than those of GCaMP6m (Fig. R4B).

And we hope the reviewer would agree with us that, the main point is DR values of NEMO (LARGER than 100) are superior over existing GECIs, not those exact measured numbers per se.

Figure R4. Background readings A) and F_{\min} -DR plots of cells with dim fluorescence B).

(C2) Under-estimation of in vitro DR

Please check your experimental conditions for in vitro measurement, if in-cell DR is found to be free from any artifacts. The in vitro performance of NEMO could be attenuated if a detector's DR and linearity were not appropriately tuned. For a PMT detector with a small DR, a signal saturation at the high-intensity range would occur to spoil the DR of NEMO. To avoid this,

authors might utilize a high-dynamic range mode of “Flexstation 3, plate reader” allowing an exceptionally wide range of intensity measurement from 10^1 to 10^6 counts. Generally, such a wide-range detection is realized by a dynamic combination of three different PMT sensitivities (high-middle-low) for which their linearity is not always guaranteed.

Response: We performed *in vitro* DR measurements of GECIs using the “high-dynamic range mode”. The results showed that the DR values of NEMO sensors were still smaller than their corresponding in-cell values (Fig. R5, Fig. R1F).

Figure R5. In vitro dynamic range of GECIs with different concentrations measured with high-dynamic range mode of Flexstation 3.

(C3) No artifacts

When authors find DR values are free from any artifacts, a logical and reasonable discussion explaining a large mismatch btw in-cell DR and in vitro DR is needed.

Response: Even with our optimized DR-measurement protocols (Fig. R1C-F), there might still be some residual calculation artifacts that cause some over-estimation of in-cell DR values, especially for those of NEMOc, NEMOf and NEMOm.

The newly obtained in-cell DR values were still larger than those *in vitro* ones. We thus added the following discussion into the main text: “It is surprising that *in vitro* DR values were smaller than their corresponding in-cell ones, as GCaMP-like design usually show the opposite (PMID:16720273). It is likely that macromolecular crowding and reducing condition in cytosolic environment may account for this higher in-cell DR (PMID: 25897121; 31994240), which warrant follow-on studies in the immediate future.”

We will investigate the underlying mechanisms in future studies. Mechanistic probing of the difference is beyond the scope of this study.

In summary,

(1) I have to say that the inconsistency of DR is unacceptable because the expanded DR is the key property of the newly generated NEMO series. (General expectation; in vitro DR>in cell DR)

Response: We agree with the reviewer that there is a discrepancy between DRs measured *in vitro* and those obtained in-cell. Nevertheless, results from both methods agreed that DR values of NEMOc/f/m/s obtained were at least 5.5 folds of GCaMP6m. Thus we hope the reviewer would agree that DR values of NEMO sensors are superior over GCaMP6 series.

We also would like to remind the reviewer about the following points:

1) Dynamic range (DR) is a non-physiological parameter. To obtain DR, non-physiological procedures are first needed to obtain maximal (F_{\max}) and minimal fluorescence (F_{\min}), then DR is calculated as $(F_{\max} - F_{\min}) / F_{\min}$. DR is only an indicator of the full capacity of signal-to-baseline-ratio (SBR) of calcium sensors. In research, SBR is the routinely used key physiological feature to describe the performance of calcium sensors, and we have shown the superior SBRs of NEMO sensors in Fig. 2 - Fig. 4.

2) Precise estimation of DR is not that necessary. As detailed in our responses, for NEMOc/f/m sensors with dramatically large DR (200~400), it is impossible to accurately measure both F_{\max} and F_{\min} without scaling. However, this is not an issue, as non-physiological reading of F_{\max} and F_{\min} are not usually required at the same time during routine physiological recordings. Researches usually utilize one portion of the full capacity of SBR, thus they only need to accurately measure of F_{\max} or F_{\min} . For example, when dealing with lower end of the DR, one can always increase the exposure time to accurately measure dim fluorescence signals; and when working within the upper end of DR, one can decrease the exposure time to avoid camera saturation.

Overall, we hope that the reviewer would agree with us that a rough estimation of the in cell DR is enough to describe the superior performance of NEMO sensors.

(2) Data normalization should be appropriately performed for Fig1C, 1D, 1F, and 1G.

Response: Un-normalized data for Fig. 1C, 1D, and 1F and 1G are now shown.

(3) For Fig1E or 1C, I believe that it is highly informative to show associated intensity data (at least in the Supplementary section) since it allows readers' at-a-glance understanding of why the DR of NEMOs is expanded.

Response: We now added these figures into supplementary figures.

Further corrections;

-DR is generally defined by $\Delta F/F_{min} = (F_{max} - F_{min})/F_{min}$, rather than F_{max}/F_{min} (line 77 and others).

Response: We have corrected these typos.

-Mixed usage of "basal" and "resting" such as " $F = (\text{basal GEC1 ratio})/(\text{rest GCaMP6m ratio})$ " at L494" is confusing. Specify their differences if these were not identical ones.

Response: We have replaced all "resting" and "rest" with "basal".

-Be consistent with the distinction between " R_0 and R_{min} " or " F_0 and F_{min} ". F_0 on the y-axis in Fig1D would be " F_{min} (F at zero- Ca^{2+})".

Response: Typo in Fig. 1D is corrected, and we have checked and made sure that R_0 and R_{min} are used correctly.

- Nuclear Export Signal is common rather than "exclusion" in Fig 1A.

Response: Corrected.

-405nm is not UV but violet light.

Response: Corrected.

-If available, cite appropriate references for plasmodesmata-associated Ca²⁺ signals to allow readers can judge whether your findings are consistent with previous reports. If this were a pretty new finding, add a reasonable explanation for this finding.

Response: We have added the following sentence into the main text : “To the best of our knowledge, this is the first report of plasmodesmata-associated Ca²⁺ signals, and opens avenues to probe the molecular machinery that govern calcium dynamics in this specialized subcellular compartment”.

-Make boxes and error bars to be clearer in Fig1F.

Response: Corrected.

-L33: Make sure the validity and consistency of DR values (>200-fold, L33 vs >100-fold, L108).

Response: Changed to >100 fold.

-L63: Update ref10 to PMID: 32571014

Response: Corrected.

Reviewer #3:

Remarks to the Author:

The authors should be applauded for their thoroughness. All reviewers' concerns have been appropriately addressed. Regarding the in vitro 2P characterization added to Figure S3, it would be nice if the authors could also provide the molecular two-photon brightness values for the examined sensors.

Response: We thank the reviewer for the supportive remark. We have followed the reviewer's advice and provided the calculated two-photon molecular brightness values of Ca²⁺-bound GECIs in the supplementary file.

Final Decision Letter:

Dear Youjun,

I am pleased to inform you that your Article, "Engineering of NEMO as calcium indicators with large dynamics and high sensitivity", has now been accepted for publication in Nature Methods. Your paper is tentatively scheduled for publication in our May print issue, and will be published online prior to that. The received and accepted dates will be June 7th, 2022 and March 16th, 2023. This note is intended to let you know what to expect from us over the next month or so, and to let you know where to address any further questions.

Once your paper is typeset, you will receive an email with a link to choose the appropriate publishing options for your paper and our Author Services team will be in touch regarding any additional information that may be required.

Please note that *Nature Methods* is a Transformative Journal (TJ). Authors may publish their research with us through the traditional subscription access route or make their paper immediately open access through payment of an article-processing charge (APC). Authors will not be required to make a final decision about access to their article until it has been accepted. [Find out more about Transformative Journals](https://www.springernature.com/gp/open-research/transformative-journals)

Authors may need to take specific actions to achieve [compliance with funder and institutional open access mandates](https://www.springernature.com/gp/open-research/funding/policy-compliance-faqs). If your research is supported by a funder that requires immediate open access (e.g. according to [Plan S principles](https://www.springernature.com/gp/open-research/plan-s-compliance)) then you should select the gold OA route, and we will direct you to the compliant route where possible. For authors selecting the subscription publication route, the journal's standard licensing terms will need to be accepted, including [self-archiving policies](https://www.springernature.com/gp/open-research/policies/journal-policies). Those licensing terms will supersede any other terms that the author or any third party may assert apply to any version of the manuscript.

Your paper will now be copyedited to ensure that it conforms to Nature Methods style. Once proofs are generated, they will be sent to you electronically and you will be asked to send a corrected version within 24 hours. It is extremely important that you let us know now whether you will be difficult to contact over the next month. If this is the case, we ask that you send us the contact information (email, phone and fax) of someone who will be able to check the proofs and deal with any last-minute problems.

If, when you receive your proof, you cannot meet the deadline, please inform us at rjsproduction@springernature.com immediately.

Once your manuscript is typeset and you have completed the appropriate grant of rights, you will receive a link to your electronic proof via email with a request to make any corrections within 48 hours. If, when you receive your proof, you cannot meet this deadline, please inform us at rjsproduction@springernature.com immediately.

Once your paper has been scheduled for online publication, the Nature press office will be in touch to confirm the details.

Once your paper has been scheduled for online publication, the Nature press office will be in touch to confirm the details.

Content is published online weekly on Mondays and Thursdays, and the embargo is set at 16:00 London time (GMT)/11:00 am US Eastern time (EST) on the day of publication. If you need to know the exact publication date or when the news embargo will be lifted, please contact our press office after you have submitted your proof corrections. Now is the time to inform your Public Relations or Press Office about your paper, as they might be interested in promoting its publication. This will allow them time to prepare an accurate and satisfactory press release. Include your manuscript tracking number NMETH-A49501B and the name of the journal, which they will need when they contact our office.

About one week before your paper is published online, we shall be distributing a press release to news organizations worldwide, which may include details of your work. We are happy for your institution or funding agency to prepare its own press release, but it must mention the embargo date and Nature Methods. Our Press Office will contact you closer to the time of publication, but if you or your Press Office have any inquiries in the meantime, please contact press@nature.com.

Nature Portfolio journals [encourage authors to share their step-by-step experimental protocols](https://www.nature.com/nature-research/editorial-policies/reporting-standards#protocols) on a protocol sharing platform of their choice. Nature Portfolio 's Protocol Exchange is a free-to-use and open resource for protocols; protocols deposited in Protocol Exchange are citable and can be linked from the published article. More details can found at www.nature.com/protocolexchange/about.

Please note that you and any of your coauthors will be able to order reprints and single copies of the issue containing your article through Nature Portfolio 's reprint website, which is located at <http://www.nature.com/reprints/author-reprints.html>. If there are any questions about reprints please send an email to author-reprints@nature.com and someone will assist you.

Best regards,
Nina

Nina Vogt, PhD
Senior Editor
Nature Methods